# RETHINKING THE BIAS OF FOUNDATION MODEL UNDER LONG-TAILED DISTRIBUTION

## ABSTRACT

Long-tailed learning has garnered increasing attention due to its practical significance. Among the various approaches, the fine-tuning paradigm has gained considerable interest with the advent of foundation models. However, most existing methods primarily focus on leveraging knowledge from these models, overlooking the inherent biases introduced by the imbalanced training data they rely on. In this paper, we examine how such imbalances affect long-tailed downstream tasks. Specifically, we refer to the biases in foundation models and downstream tasks as parameter imbalance and data imbalance, respectively. Through fine-tuning, we observe that parameter imbalance plays a more critical role, while data imbalance can be mitigated using existing re-balancing strategies. Moreover, we find that parameter imbalance cannot be effectively addressed by current re-balancing techniques, such as adjusting the logits, during training, unlike data imbalance. To tackle both imbalances simultaneously, we constitute a causal structure graph and view the incomplete semantic factor as the confounder, which brings spurious correlations between input samples and labels. To resolve the negative effects of this, we propose a novel backdoor adjustment method that learns the true causal effect between input samples and labels, rather than merely fitting the correlations in the data. Experimental results validate the effectiveness of our method.

## 1 INTRODUCTION

Real-world data often follows a long-tailed distribution, where the majority of instances are concentrated in head classes, leaving only a small number of instances for each tail class. This scarcity of samples makes generalization on such labels challenging, and naive learning on this data tends to introduce an undesirable bias toward dominant labels. Recently, with the advent of foundation models, downstream performance can be significantly improved by fine-tuning these models on labeled data, often yielding superior results compared to training from scratch, while also minimizing training costs (Wang et al., 2022b;a). Consequently, there has been considerable interest in exploring how to fine-tune foundation models and leverage their strong generalization capabilities to enhance learning for tail classes.

Recent works such as LIFT (Shi et al., 2024), LPT (Dong et al., 2022), and VL-LTR (Tian et al., 2022) demonstrate that properly fine-tuning foundation models like CLIP (Radford et al., 2021) can significantly enhance long-tail learning performance. VL-LTR improves visual recognition, particularly for tail classes, by collecting class descriptions from the internet and jointly learning both visual and text representations. LIFT, on the other hand, reveals that heavy fine-tuning hurts and Parameter-Efficient Fine-Tuning (Chen et al., 2022; Jia et al., 2022) (PEFT) based methods can preserve as much information from the foundation model as possible, which is important for the downstream imbalanced learning. However, these methods tend to overemphasize the use of foundation models while overlooking their inherent biases, as shown in Fig. 1. Large-scale datasets used to train foundation models, such as LAION (Schuhmann et al., 2021), also follow a long-tailed distribution, which can negatively impact downstream tasks (Zhu et al., 2024; Wen et al., 2024). Therefore, the fine-tuned model is influenced by dual long-tailed distributions (upstream and downstream imbalance ), and only considering data imbalance is not proper.

In this paper, we explore *how the imbalance of foundation models impacts downstream imbalanced tasks in PEFT-based methods*. Since the pre-training data is inaccessible, its influence is primarily reflected in the pre-trained weights, or parameters, which we refer to as **parameter imbalance**. In contrast, downstream data is accessible and directly affects the downstream task, which we define as **data imbalance**. Through fine-tuning, we find that both types of imbalance influence downstream tasks, but parameter imbalance

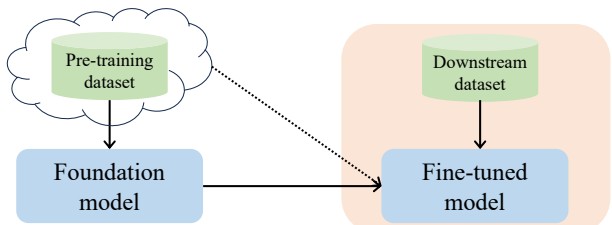

Figure 1: Previous methods focus on how to use the downstream data to fine-tune the foundation model while ignoring that the pre-training data has a potential influence (dashed line).

plays a more significant role, as shown in Fig. 2 and Fig. 3. In addition, we locate that samples belonging to the tail classes grouped by the data and parameter imbalance at the same time are influenced extremely. Due to the inaccessibility of pre-training data, we estimate the label prior and extend the current Generalized Logit Adjustment (GLA) (Zhu et al., 2024) into the training phase and named the method GLA-Train. Unlike Logit Adjustment (Menon et al., 2020)(LA), which effectively addresses data imbalance, we find that GLA-Train cannot be used to alleviate parameter imbalance, as shown in Tab. 3. After measuring the quality of feature representation via K-Nearest-Neighbor (KNN) (Cover & Hart, 1967) accuracy, we surprisingly find that LA slightly enhances the feature representation of tail classes and the improvement mostly comes from the classifier, as shown in Tab. 4. When the imbalance is integrated into the parameter, it is difficult to relive it via adjusting the logit. Therefore, we summarize that parameter imbalance is essentially different from the data imbalance and we cannot simply resolve it by re-balance based method.

To tackle both parameter and data imbalances simultaneously, we build a causal structure graph and find that the incomplete semantic factor is the confounder, encouraging the model to learn spurious correlations between input samples and labels and restricting its generalization ability. For instance, if the class "dog" belongs to the tail classes due to parameter imbalance, the foundation model may lack sufficient semantic information, causing it to capture only partial features, such as the dog's head, to represent the class. When this model is adapted to a downstream task where "dog" is also a tail class due to data imbalance, fine-tuning struggles to learn the complete set of relevant semantic features, further limiting its generalization. We denote the dog's head as a typical example of a incomplete semantic factor. To inhibit the confounding effect, we adopt the backdoor criterion in causal inference to realize our backdoor adjustment. After applying our method, we achieve a more balanced performance over all the classes on different datasets.

Our contributions are summarized as follows:

- We consider a practical problem in that the data to train the foundation model and adapt to the downstream task are both imbalanced, which is denoted by parameter imbalance and data imbalance, respectively. After fine-tuning, we find that the parameter imbalance plays a more important role than the data imbalance and current re-balancing based methods cannot handle it effectively. Moreover, we find that trying to re-balance the parameter imbalance can not give much benefit to the representation.

- We find that the incomplete semantic factor encourages the model to learn spurious correlations between input samples and labels, which restricts its generalization ability. To solve that, we construct a causal graph and propose a backdoor adjustment method to eliminate the confounder negative impact.

- We conduct extensive experiments on the ImageNet-LT (Deng et al., 2009), Places365-LT (Liu et al., 2019), and iNaturalist2018 (Van Horn et al., 2018) datasets, and the results verify the effectiveness of our method.

## 2 RELATED WORK

**Long-tailed learning** Previous methods primarily focus on re-balancing tail classes, employing techniques such as re-weighting (Cui et al., 2019) or re-sampling (Ren et al., 2020; Guo & Wang,

Table 1: The Zero-Shot (ZS) performance of different type of CLIPs on different benchmarks.

| | ImageNet-LT | | | Places365-LT | | | iNaturalist2018 | | |
|---|---|---|---|---|---|---|---|---|---|
| | D-Many | D-Medium | D-Few | D-Many | D-Medium | D-Few | D-Many | D-Medium | D-Few |
| CLIP | 67.87 | 66.27 | 66.03 | 36.59 | 38.01 | 45.53 | 3.60 | 4.38 | 4.10 |
| OpenCLIP | 67.15 | 65.02 | 64.31 | 41.40 | 38.06 | 40.28 | 2.45 | 2.80 | 2.38 |
| MetaCLIP | 70.98 | 68.70 | 68.57 | 38.79 | 37.93 | 40.44 | 5.07 | 6.37 | 6.00 |

2021; Kim et al., 2020). The fundamental concept behind these methods is to place greater emphasis on tail classes to alleviate the effects of imbalanced bias. For instance, in (Cui et al., 2019), different classes are assigned a weight based on the effective number of samples in the final loss function. Similarly, Logit adjustment (Menon et al., 2020) re-balances tail classes by adjusting the output logits according to the prior for each class. Recently, foundation models, which are pre-trained on vast amounts of curated data, have demonstrated their utility in producing generalizable features transferable across various tasks. Several approaches have integrated foundation models to address long-tailed learning (Dong et al., 2022; Shi et al., 2024; Tian et al., 2022; Ma et al., 2021). However, these methods typically only consider the data distribution bias in downstream tasks, overlooking the inherent imbalance within foundation models themselves (Zhu et al., 2024). In this work, we investigate the compound effects of both pre-trained and downstream imbalances, and we propose a simple approach to mitigate these effects.

**Downstream fine-tuning**   Fine-tuning techniques can be broadly categorized into full fine-tuning and Parameter-Efficient Fine-Tuning (PEFT) (Jia et al., 2022; Chen et al., 2022; Zaken et al., 2021; Houlsby et al., 2019; Hu et al., 2021). PEFT refers to methods that adapt pre-trained models to specific tasks while minimizing the number of parameters that require updating. When data samples are limited, PEFT often outperforms full fine-tuning. Visual Prompt Tuning (VPT) (Jia et al., 2022) introduces two variants, VPT-Shallow and VPT-Deep, which insert prompts into different transformer layers. In contrast, AdaptFormer (Chen et al., 2022) introduces lightweight modules that add only a small number of parameters, yet outperform fully fine-tuned models on various benchmarks. Despite their efficiency, PEFT methods focus on introducing fewer parameters while preserving the foundational model's information. However, this also retains the model's inherent limitations, such as imbalances. In this work, we provide an in-depth analysis of how the biases of foundation models impact downstream tasks and propose a novel method that achieves promising performance.

## 3   PRELIMINARY

**Notation**   We aim to solve a $C$-way classification problem with instances $\boldsymbol{x} \in \mathcal{X}$ and labels $y \in \mathcal{Y} = [C] = \{1, \ldots, C\}$, where $\mathcal{X}$ and $\mathcal{Y}$ denote the input space and output space. For pre-training, we denote $\mathcal{D}_P$ as the distribution of the training set and we cannot access it at the fine-tuning phase. For downstream tasks, we denote $\mathcal{D}_S$ and $\mathcal{D}_T$ as the distribution for training and test, respectively. In the context of long-tailed learning, the number of samples varies across classes, i.e., $\mathbb{P}_S(Y = 1) \neq \mathbb{P}_S(Y = 2) \neq \ldots \neq \mathbb{P}_S(Y = C)$, where $\mathbb{P}_S(Y)$ denotes the class prior of $\mathcal{D}_S$. In contrast, the test set $T$ is sampled from the distribution $\mathcal{D}_T$, where each class $c$ has an equal probability, i.e., $\mathbb{P}_T(Y = c) = 1/C$. Our target is to learn a hypothesis $f : \mathcal{X} \rightarrow \mathcal{Y}$ to estimate the posterior probability $\mathbb{P}_S(Y = y \mid \boldsymbol{x})$ from the training set and generalize it to the test set.

**Logit adjustment (LA)**   Under the label shift assumption, we have $\mathbb{P}_S(\boldsymbol{x} \mid Y = y) = \mathbb{P}_T(\boldsymbol{x} \mid Y = y)$ but $\mathbb{P}_S(Y = y \mid \boldsymbol{x}) \neq \mathbb{P}_T(Y = y \mid \boldsymbol{x})$ for each class $y$. LA (Menon et al., 2020) bridges the gap between the posterior of the imbalanced training set and the balanced test set. As shown in Eq. 1, we can get the posterior of the test set from the training set by introducing a scaling factor.

$$
\begin{aligned}
\mathbb{P}_T(Y = y \mid \boldsymbol{x}) &= \frac{\mathbb{P}_T(\boldsymbol{x} \mid Y = y)\mathbb{P}_T(Y = y)}{\mathbb{P}_T(\boldsymbol{x})} \\
&\propto \mathbb{P}_S(\boldsymbol{x} \mid Y = y)\mathbb{P}_T(Y = y) \propto \frac{\mathbb{P}_T(Y = y)}{\mathbb{P}_S(Y = y)}\mathbb{P}_S(Y = y \mid \boldsymbol{x})
\end{aligned}
\tag{1}
$$

There are two types of LA, either applied post-hoc to a trained model or enforced in the loss during training, and the latter can achieve better performance. When integrating it to the criterion, we get the final LA loss, as shown in Eq. 2, where $f_y(\boldsymbol{x} \mid \boldsymbol{\theta}, \boldsymbol{\phi}) \propto \mathbb{P}(Y = y \mid \boldsymbol{x})$, $\boldsymbol{\theta}$, and $\boldsymbol{\phi}$ denote

|  | 75.05 | 53.06 | 34.31 |
| P-Medium P-Many | 58.10 | 35.93 | 21.13 |
| P-Few | 40.72 | 25.36 | 10.56 |
|  | D-Many | D-Medium | D-Few |

(a) CE

|  | 72.51 | 66.94 | 63.34 |
| P-Medium P-Many | 49.95 | 50.88 | 47.35 |
| P-Few | 35.78 | 39.87 | 31.12 |
|  | D-Many | D-Medium | D-Few |

(b) LA

Figure 2: The performance of different groups with (a) CE and (b) LA on Places365-LT dataset. The performance of the row of "P-Few" is terrible.

Table 2: The Average Accuracy Gap between different foundation models on Places365-LT with ZS, CE, and LA.

| $f - g$ | ZS | CE | LA |
|---|---|---|---|
|  |  | Adaptformer | |
| CLIP−OpenCLIP | 10.38 | 3.19 | 3.49 |
| CLIP−MetaCLIP | 11.56 | 3.22 | 3.32 |
| OpenCLIP−MetaCLIP | 12.03 | 3.03 | 3.40 |
|  |  | VPT | |
| CLIP−OpenCLIP | 10.38 | 3.6 | 3.94 |
| CLIP−MetaCLIP | 11.56 | 3.42 | 3.87 |
| OpenCLIP−MetaCLIP | 12.03 | 3.10 | 3.53 |

the output logit of class $y$, the parameter of the foundation model, and the additional fine-tuned parameter, respectively. If not specified, we freeze the $\boldsymbol{\theta}$ and only optimize the $\boldsymbol{\phi}$.

$$L_{LA}\big(f(\boldsymbol{x} \mid \boldsymbol{\theta}, \boldsymbol{\phi}), y\big) = \log \Big[1 + \sum_{y' \neq y} \Big(\frac{\mathbb{P}_S(Y = y')}{\mathbb{P}_S(Y = y)}\Big) \cdot e^{f_{y'}(\boldsymbol{x}|\boldsymbol{\theta},\boldsymbol{\phi}) - f_y(\boldsymbol{x}|\boldsymbol{\theta},\boldsymbol{\phi})}\Big] \quad (2)$$

**Generalized logit adjustment (GLA)** Since the label prior $\mathbb{P}_P(Y = y)$ of the pre-training data is inaccessible and we cannot use Eq. 1 to estimate the posterior (from $\mathbb{P}_S(\boldsymbol{x} \mid Y = y)$, $\mathbb{P}_S(Y = y)$, and $\mathbb{P}_S(Y = y \mid \boldsymbol{x})$ to $\mathbb{P}_P(\boldsymbol{x} \mid Y = y)$, $\mathbb{P}_P(Y = y)$, and $\mathbb{P}_P(Y = y \mid \boldsymbol{x})$ ). GLA (Zhu et al., 2024) estimates the prior following the Eq. 3 on the validation set, where $L_{CE}$ denotes the Cross-Entropy (CE) loss. After getting the estimated $\widehat{\mathbb{P}}_P(Y)$, we can adjust the logit following Eq. 1.

$$\widehat{\mathbb{P}}_P(Y) = \min_{\boldsymbol{q}} \max_{\lambda \geq 0, v} \mathbb{E}_{(\boldsymbol{x},y) \sim \mathcal{D}_T} L_{CE}(f(\boldsymbol{x} \mid \boldsymbol{\theta}) - \log \boldsymbol{q}, y) - \sum_i \lambda_i \boldsymbol{q}_i + v(1 - \sum_{i \in [C]} \boldsymbol{q}_i) \quad (3)$$

GLA assumes that the Zero-Shot (ZS) and Fine-Tuned (FT) models have diverse predicitons (Zhu et al., 2024). In this way, we can achieve unbiased predictions by simply ensembling the output logit of two models, as shown in Eq. 4, where $f(\boldsymbol{x} \mid \boldsymbol{\theta})$ and $f(\boldsymbol{x} \mid \boldsymbol{\theta}, \boldsymbol{\phi})$ denote the output logit of ZS and FT, respectively. Since the adjustments of the two models are individual, we split Eq. 4 into GLA-ZS and GLA-FT, eliminating the bias of the foundation model and downstream data, respectively.

$$f'_y(\boldsymbol{x}) = \underbrace{f_y(\boldsymbol{x} \mid \boldsymbol{\theta}) - \log \widehat{\mathbb{P}}_P(Y = y)}_{GLA-ZS} + \underbrace{f_y(\boldsymbol{x} \mid \boldsymbol{\theta}, \boldsymbol{\phi}) - \log \mathbb{P}_S(Y = y)}_{GLA-FT} \quad (4)$$

## 4 WHEN THE DATA IMBALANCE MEETS THE PARAMETER IMBALANCE

### 4.1 PARAMETER IMBALANCE IS MORE IMPORTANT

We consider the problem of the pre-training data and the downstream training data being both imbalanced. For the parameter imbalance, we give theoretical definition in Def. 1. Since the $\mathbb{P}_P(Y)$ is not accessible, we use the estimated prior $\widehat{\mathbb{P}}_P(Y)$ to substitute $\mathbb{P}_P(Y)$ for the following analysis.

**Definition 1** *Let $\mathbb{P}_P(Y)$ denote the label of the pre-training data. We have $\mathbb{P}_P(Y = 1) \neq \mathbb{P}_P(Y = 2) \neq \ldots \neq \mathbb{P}_P(Y = C)$. We use the imbalanced factor (IF) IF $= \max_{c \in [C]} \mathbb{P}_P(Y) / \min_{c \in [C]} \mathbb{P}_P(Y)$ to measure the degree of parameter imbalance.*

Most previous works (Dong et al., 2022; Shi et al., 2024; Tian et al., 2022) focus on eliminating the data bias to improve the performance but ignore the parameter imbalance. In the following, we will explore the impact of the superposition of two imbalances.

**Experiment setup** We conduct experiments with ViT-B/16 (Dosovitskiy et al., 2020) on the ImageNet-LT, the Places365-LT, and iNaturalist2018 datasets. To better explore the influence of

pre-trained data, we select CLIP (Radford et al., 2021), OpenCLIP (Cherti et al., 2023), and Meta-CLIP (Xu et al., 2023a) as foundation models, which are pre-trained on WIT, LAION (Schuhmann et al., 2021), and MetaData, respectively. Previous works (Shi et al., 2024; Dong et al., 2022) find that we can achieve promising performance by fine-tuning only a small proportion of parameters. Therefore, we select two typical PEFT techniques, Adaptformer (Chen et al., 2022) and Visual Prompt Tuning (VPT) (Jia et al., 2022), as the basic methods. The learning rate, number of epochs, and parameter initialization strategies follows (Shi et al., 2024). Following OLTR (Liu et al., 2019), we split the classes into three groups named "D-Many", "D-Medium", and "D-Few" relying on the number of samples. Similarly, for parameter imbalance, we split the classes into three groups named "P-Many", "P-Medium", and "P-Few" relying on $\widehat{\mathbb{P}}_P(Y)$. More details are in the Appendix Sec. A.

**Different foundation models serve different parameter imbalance**  We report the ZS performance on different long-tailed datasets, as shown in Tab. 1. Since the imbalanced downstream dataset does not influence the foundation models, it is natural that we cannot observe the data imbalance, i.e., "D-Many" and "D-Few" should have a similar performance. However, on the Places365-LT, "D-Few" is better than "D-Many" for the CLIP model and other foundation models are nearly similar. Since they all share the architecture, the main differences come from the imbalance pre-training data, i.e., parameter imbalance. To better measure the difference, we define Average Accuracy Gap $\Delta_{avg}^{f,g}$, where $f, g$ and $Acc(f, c)$ denote two arbitrary hypotheses and the $c$-th class-wise accuracy of model $f$, respectively.

$$\Delta_{avg}^{f,g} = \frac{1}{C} \sum_{c=1}^{C} |Acc(f,c) - Acc(g,c)| \tag{5}$$

As shown in Tab. 2, different foundations have merely 10% differences on the same downstream task. When we fine-tune them to adapt the downstream, the gap is decreased but still exists.

**The parameter imbalance occupys**
When we adapt the foundation model to the downstream task, the fine-tuned model is influenced by parameter and data imbalance. For analysis, we conduct experiments with different criterions, CE and LA, on the CLIP model. As shown in Fig. 3, CE suffers from the data imbalance heavily although we adapt from the foundation model, which is not influenced by it. When changing the criterion from CE to LA, the data imbalance is relieved effectively and we achieve a more balanced performance. As for parameter imbalance, although the fine-tuning-based method can alleviate it, the bias still exists. We primarily give two explanations for this phenomenon. One is that current criteria do not consider the parameter imbalance explicitly. Another is that the PEFT method fixes the parameters of the foundation model, while retaining strong generalization capabilities, the parameter imbalance is also preserved.

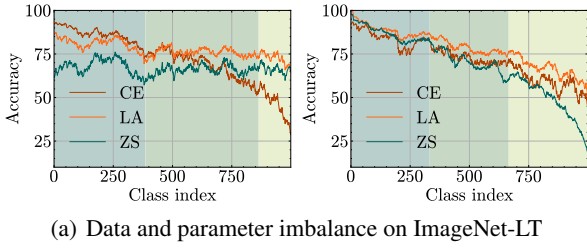

(a) Data and parameter imbalance on ImageNet-LT

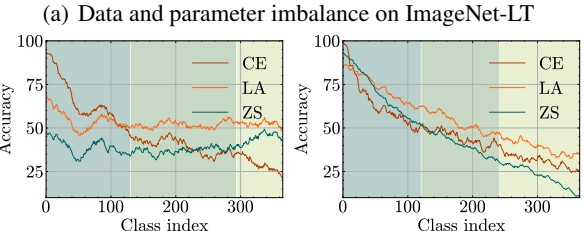

(b) Data and parameter imbalance on Places365-LT

Figure 3: The data and parameter imbalance on (a) ImageNet-LT and (b) Places365-LT. The class indices of the left picture are sorted relying on the data imbalance while the right picture relies on the parameter imbalance. Curves are smoothed for better visualization. Parameter imbalance occupies a more vital role.

As for data imbalance, only a small proportion of parameters is influenced by it, and the re-balancing technique, i.e. LA, can easily eliminate it. Therefore, parameter imbalance plays a more vital role than data imbalance.

**Tail-Tail hurts**  Parameter imbalance is a critical factor that impedes further improvement. To analyze this, we consider two types of imbalances in conjunction and divide the validation sets

Table 3: Ther performance of GLA, GLA-ZS, GLA-FT, and GLA-Train based on the CLIP model.

|  | Type | D-Many | D-Medium | D-Few | P-Many | P-Medium | P-Few | Overall |
|---|---|---|---|---|---|---|---|---|
| ImageNet-LT | GLA | 79.76 | 76.48 | 74.12 | 88.34 | 78.58 | 65.30 | 77.42 |
|  | GLA-ZS | 70.23 | 68.61 | 68.90 | 83.63 | 70.36 | 53.79 | 69.27 |
|  | GLA-FT | 79.67 | 76.15 | 73.29 | 88.07 | 77.92 | 65.32 | 77.12 |
|  | GLA-Train | 80.21 | 75.93 | 72.01 | 88.05 | 77.93 | 65.32 | 77.10 |
| Places365-LT | GLA | 51.22 | 52.00 | 53.34 | 69.07 | 50.70 | 35.87 | 51.98 |
|  | GLA-ZS | 40.58 | 39.57 | 46.63 | 61.33 | 39.69 | 22.56 | 41.31 |
|  | GLA-FT | 50.41 | 52.23 | 52.18 | 67.71 | 49.57 | 37.15 | 51.56 |
|  | GLA-Train | 51.36 | 52.30 | 50.91 | 67.70 | 49.58 | 37.17 | 51.48 |

into nine groups. For instance, some classes can be categorized as "D-Many" due to data imbalance, while simultaneously classified as "P-Few" because of parameter imbalance. As shown in Fig. 2, we observe that classes falling into both "D-Few" and "P-Few" classes exhibit the poorest performance. Moreover, when we shift the criterion from CE to LA, we find that the performance of "P-Few" classes within the "D-Many" group deteriorates. This indicates that previous claims suggesting LA can alleviate parameter imbalance do so at the expense of "P-Few" class performance, failing to address the parameter imbalance fundamentally.

## 4.2 ANALYSIS OF ADJUSTMENT FOR PARAMETER IMBALANCE

Addressing parameter imbalance is a crucial task. Drawing inspiration from methods used to tackle data imbalance, a natural approach to mitigating this issue is to adjust the output logits, similar to the Logit Adjustment (LA) technique. The existing Generalized Logit Adjustment (GLA) method addresses parameter imbalance by modifying the logits according to the estimated label priors. Thus, in this section, we explore whether parameter imbalance can be effectively eliminated through the simple adjustment of output logits.

**GLA fails during training** Firstly, We follow GLA to estimate the prior of the pre-training data and then adjust the output logit following Eq. 4. As shown in Tab. 3, we achieve a more balanced performance and verify its effectiveness. However, GLA achieves unbiased prediction by modeling the parameter imbalance and data imbalance respectively, which is more complicated. Referring to LA, which models the adjustment into the criterion, a natural improvement is extending GLA into training. This approach mitigates parameter bias by introducing additional unbiased parameters. We denote this method as GLA-Train and constitute the optimization target as shown in Eq. 2. Since the parameter imbalance does not influence the classifier, we only use Eq. 6 to learn the unbiased representation and introduce an additional stage for classifier re-training (Kang et al., 2019).

$$L_{GLA}\big(f(\boldsymbol{x} \mid \boldsymbol{\theta}, \boldsymbol{\phi}), y\big) = \log\Big[1 + \sum_{y' \neq y}\Big(\frac{\mathbb{P}_S(Y = y')\widehat{\mathbb{P}}_P(Y = y')}{\mathbb{P}_S(Y = y)\widehat{\mathbb{P}}_P(Y = y)}\Big) \cdot e^{f_{y'}(\boldsymbol{x}\mid\boldsymbol{\theta},\boldsymbol{\phi}) - f_y(\boldsymbol{x}\mid\boldsymbol{\theta},\boldsymbol{\phi})}\Big] \quad (6)$$

The results are presented in Tab.3. In comparison with GLA, GLA-Train offers minimal additional benefit in addressing both parameter and data imbalance, achieving a performance similar to that of LA, as shown in Fig.2. Since Eq. 6 explicitly models both data and parameter imbalance, it is important to note that only the data imbalance is effectively corrected while parameter imbalance is ignored. This indicates that parameter imbalance is fundamentally different from data imbalance and cannot be resolved through simple adjustment alone.

**Feature representation analysis** To explore how the re-balance influences the feature representation, we utilize the balanced validation sets of ImageNet-LT and Places365-LT to assess the feature quality of the test set through KNN accuracy. As presented in Tab. 4, we observe that all three methods yield comparable accuracy. The re-balancing-based methods provide slight improvements for the "D-Few" and "D-Medium" classes, but this comes at the expense of performance in

Table 4: The KNN accuracy. We denote GLA-T as the GLA-Train.

|  |  | D-Many | D-Medium | D-Few | All |
|---|---|---|---|---|---|
| ImageNet-LT | CE | 74.61 | 70.02 | 66.14 | 71.25 |
|  | LA | 74.42 | 70.35 | 66.43 | 71.36 |
|  | GLA-T | 74.33 | 70.35 | 66.31 | 71.26 |
| Places365-LT | CE | 44.53 | 46.01 | 47.92 | 45.85 |
|  | LA | 44.14 | 46.23 | 48.75 | 45.93 |
|  | GLA-T | 44.33 | 46.14 | 48.66 | 45.93 |

ments for the "D-Few" and "D-Medium" classes, but this comes at the expense of performance in

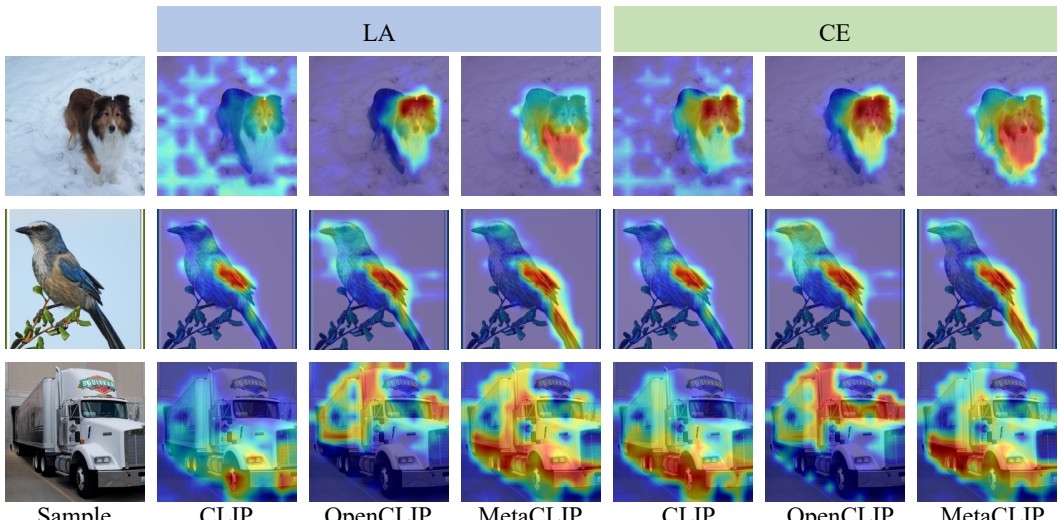

|     | LA |  |  | CE |  |  |
| Sample | CLIP | OpenCLIP | MetaCLIP | CLIP | OpenCLIP | MetaCLIP |

Figure 4: We randomly select three samples of different tail classes and visualize the heatmap via Grad-CAM (Selvaraju et al., 2017) with CE and LA. Different models draw attention to different areas and merging them can acquire unbiased semantic information.

the "D-Many" class. This suggests that the enhanced performance for tail classes can be attributed to a more accurate classifier, which can be achieved by employing re-balancing techniques like Logit Adjustment (LA). We also conduct experiments to explore the influence of classifier in the Appendix Sec. D.2. However, addressing parameter imbalance—manifested in the parameters of foundation models—proves to be more challenging. It is difficult to mitigate the negative effects of parameter imbalance solely through the classifier. Consequently, re-balancing methods cannot effectively alleviate parameter imbalance.

## 5 METHOD

As previously analyzed, the parameter imbalance becomes dominant after adapting to the downstream task, and we cannot eliminate it solely by adjusting the logits during the training phase. Therefore, exploring an effective method to address both parameter imbalance and data imbalance simultaneously is essential. In this section, instead of focusing on re-balancing methods, we analyze the issue from a causal perspective. By identifying the latent confounder, we can apply the backdoor criterion to estimate its negative impact, thereby achieving a more balanced performance across various classes.

### 5.1 A CAUSAL VIEW FOR LONG-TAILED LEARNING

We model the long-tailed image classification process with a causal structure graph as shown in Fig. 5. Here, we denote $X$, $C$, $U$, and $D$ as an imbalanced dataset, the incomplete semantic factor, the inaccessible semantic factor, and the parameter imbalance, respectively. $B$ is the data balanced representation and $Y$ is the predicted label distribution. In Fig. 5, $X \to B$ denotes we extract the data balanced representation from an imbalanced dataset. We can achieve this by applying re-balancing based method like LA. $B \to Y$ denotes the inference pipeline that we predict the label relying on the given representation. $D \to C$ presents the incomplete semantic factor depending on the parameter imbalance. From the perspective of data generation, $U \to X \leftarrow C$ denotes the

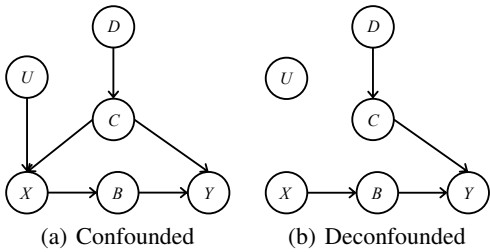

(a) Confounded          (b) Deconfounded

Figure 5: The framework of our proposed method. We view parameter imbalance as the confounder and remove its negative impact.

Table 5: Results on Places365-LT.

| Method | Backbone | Extra data | Params. | Epochs | D-Many | D-Medium | D-Few | All |
|--------|----------|-----------|---------|--------|--------|----------|-------|-----|
| LiVT | ViT-B/16 | ✗ | 85.80M | 100 | 48.1 | 40.6 | 27.5 | 40.8 |
| LPT | ViT-B/16 | ✗ | 1.01M | 80 | 49.3 | 52.3 | 46.9 | 50.1 |
| VL-LTR | ViT-B/16 | ✔ | 149.62M | 100 | **54.2** | 48.5 | 42.0 | 50.1 |
| RAC | ViT-B/16 | ✔ | 85.80M | 30 | 48.7 | 48.3 | 41.8 | 47.2 |
| BALLAD | ViT-B/16 | ✔ | 149.62M | 60 | 49.3 | 50.2 | 48.4 | 49.5 |
| Decoder | ViT-B/16 | ✗ | 21.26 | 34 | - | - | - | 46.8 |
| LIFT | ViT-B/16 | ✗ | 0.18M | 20 | 51.3 | 52.2 | 50.5 | 51.5 |
| Ours | ViT-B/16 | ✗ | 0.54M | 10 | 52.52 | **53.62** | **52.54** | **53.01** |

dataset can be generated by giving the accessible semantic factor and inaccessible semantic factor, i.e., $P(X) = \sum P(X|C,U)P(C)P(U)$. $C \rightarrow Y$ indicates that the predicted label distributions follow their own preferences for incomplete semantic factor. We also give additional experimental evidence for $C \rightarrow Y$ in Sec. D.7.

Therefore, the incomplete semantic factor $C$ in our problem setting, acting as a confounder (Pearl, 2009), can create a backdoor path $X \leftarrow C \rightarrow Y$, leading to spurious correlation between $X$ and $Y$. If we ignore the influence of incomplete semantic factor and learn the posterior $\mathbb{P}_S(Y = y \mid \boldsymbol{x})$ to estimate $\mathbb{P}_T(Y = y \mid \boldsymbol{x})$, the spurious correlation will be modeled, which leads to a biased model. As shown in Fig. 4, we visualize given samples on different fine-tuned models, where the corresponding foundation models serve different parameter imbalances, i.e. incomplete semantic factor. For the specific picture, different models are drawn in different interesting areas. The relationship between the confounding path $X \leftarrow C \rightarrow Y$ is unstable. For example, in the first row of Fig. 4, OpenCLIP is more attracted by the head while MetaCLIP is more interested in the body of the dog. Therefore, if a test sample belonging to this class but the body is obscured, MetaCLIP is easier to give the wrong prediction. We analyze that each incomplete semantic factor can provide sufficient information to distinguish the given class from others on the training set. However, it also indicates that the incomplete semantic factor can prohibit the model from learning other relevant information, which limits its generalization. Therefore, eliminating the influence of incomplete semantic factor is vital for imbalanced learning.

## 5.2 Backdoor adjustment

A more generalized fine-tuned model should be independent of incomplete semantic factor (i.e. parameter imbalance), which inhibits confounding effects from $C$. Therefore, instead of estimating $\mathbb{P}(Y = y \mid \boldsymbol{x})$, we use back-door criterion and estimate $\mathbb{P}(Y = y \mid do(\boldsymbol{x}))$, where $do()$ is exploited to cut off the connection from the $C$ to $X$. Considering that each sample in $X$ uniquely corresponds to a balanced representation in $B$, which indicates that the mapping between $X$ and $B$ is injective. Thus there only exists a certain $b$ such that $\mathbb{P}(\boldsymbol{b} \mid \boldsymbol{x}) = 1$ and $\mathbb{P}(\boldsymbol{b}' \mid \boldsymbol{x}) = 0$ for $\boldsymbol{b} \neq \boldsymbol{b}'$. Then, we propose our back-door adjustment method:

$$\mathbb{P}(Y = y \mid do(\boldsymbol{x})) = \sum_{\boldsymbol{b} \in B} \sum_{c \in C} \mathbb{P}(Y = y \mid \boldsymbol{b}, c)\mathbb{P}(c)\mathbb{P}(\boldsymbol{b} \mid \boldsymbol{x}) = \sum_{c \in C} \mathbb{P}(Y = y \mid \boldsymbol{b}, c)\mathbb{P}(c) \tag{7}$$

For simplicity, we assume $\mathbb{P}(c) = 1/M$, where $M$ is the number of incomplete semantic factors. Intuitively, $\mathbb{P}(Y = y \mid do(\boldsymbol{x}))$ can be estimated by fusing $\mathbb{P}(Y = y \mid \boldsymbol{b}, c)$, where $\boldsymbol{b}$ represents the balanced representation. For different incomplete semantic factors $\{c_1, c_2, \cdots, c_M\}$, we utilize a re-balancing method, such as Logit Adjustment (LA), to compute a set of data balanced outputs $\{\mathbb{P}(Y = y \mid \boldsymbol{b}, c_1), \mathbb{P}(Y = y \mid \boldsymbol{b}, c_2), \cdots, \mathbb{P}(Y = y \mid \boldsymbol{b}, c_M)\}$. These outputs can then be merged using fusion weights, following Eq. 7, to obtain an unbiased estimation. Since iterating over all possible incomplete semantic factors is impractical, we approximate these factors using models like CLIP, OpenCLIP, and MetaCLIP, thereby addressing Eq. 7 and simplifying the process of balancing both parameter and data imbalances simultaneously.

## 6 Experiment

We conduct experiments on the ImageNet-LT, Places365-LT, and iNaturalist2018 datasets. Most of our experimental setup follows previous descriptions in Sec. 4.1. For a fair comparison, we only

Table 6: Results on ImageNet-LT.

| Method | Backbone | Extra data | Params. | Epochs | D-Many | D-Medium | D-Few | All |
|---|---|---|---|---|---|---|---|---|
| LiVT | ViT-B/16 | ✘ | 85.80M | 100 | 73.6 | 56.4 | 41.0 | 60.9 |
| VL-LTR | ViT-B/16 | ✔ | 149.62M | 100 | **84.5** | 74.6 | 59.3 | 77.2 |
| BALLAD | ViT-B/16 | ✔ | 149.62M | 60 | 79.1 | 74.5 | 69.8 | 75.7 |
| Decoder | ViT-B/16 | ✘ | 21.26M | 18 | - | - | - | 73.2 |
| GML | ViT-B/16 | ✔ | 149.62M | 100 | - | - | - | 78.0 |
| LIFT | ViT-B/16 | ✘ | 0.18M | 20 | 80.2 | 76.1 | 71.5 | 77.0 |
| Ours | ViT-B/16 | ✘ | 0.54M | 20 | 82.21 | **78.75** | **74.99** | **79.57** |

Table 7: Results on iNaturalist2018.

| Method | Backbone | Extra data | Params. | Epochs | D-Many | D-Medium | D-Few | All |
|---|---|---|---|---|---|---|---|---|
| LiVT | ViT-B/16 | ✘ | 85.80M | 100 | 78.9 | 76.5 | 74.8 | 76.1 |
| LPT | ViT-B/16 | ✘ | 1.01M | 160 | - | - | 79.3 | 76.1 |
| VL-LTR | ViT-B/16 | ✔ | 149.62M | 100 | - | - | - | 76.8 |
| RAC | ViT-B/16 | ✔ | 85.80M | 20 | 75.9 | 80.5 | 81.1 | 80.2 |
| Decoder | ViT-B/16 | ✘ | 21.26M | 5 | - | - | - | 59.2 |
| LIFT | ViT-B/16 | ✘ | 4.75M | 20 | 72.4 | 79.0 | 81.1 | 79.1 |
| Ours | ViT-B/16 | ✘ | 14.25M | 20 | **82.30** | **81.67** | **82.36** | **82.01** |

compare our method with LiVT (Xu et al., 2023b), LPT (Dong et al., 2022), VL-LTR (Tian et al., 2022), RAC (Long et al., 2022), LIFT (Shi et al., 2023), Decoder (Wang et al., 2024), GML (Suh & Seo, 2023) and BALLAD (Ma et al., 2021), which are trained based on the ViT (Dosovitskiy et al., 2020). If not specified, We use the Adaptformer to fine-tune the foundation model (We also report the results based on VPT in the Appendix). More details are in the Appendix Sec. A.

## 6.1 RESULTS

**Places365-LT** The results are presented in Tab. 5, where our method demonstrates superior overall performance compared to other approaches. Specifically, in comparison to VL-LTR, which leverages additional data for representation learning, our method achieves an overall performance improvement of $2.91\%$. It is worth noting that while VL-LTR achieves the highest performance in the "D-Many" group, this comes at the cost of reduced performance in the "D-Medium" and "D-Few" groups. In contrast, our method delivers consistently higher and more balanced performance across all groups, indicating its effectiveness in addressing long-tailed distributions.

**ImageNet-LT** The results in Tab. 6 demonstrate that our method consistently improves performance across all categories. Specifically, our approach provides gains of $2.01\%$, $2.65\%$, and $3.49\%$ for the "D-Many", "D-Medium", and "D-Few" classes, respectively, compared to LIFT. Notably, the largest improvements are observed in the tail classes, underscoring the significance of our back-door adjustment technique in addressing challenges faced by these underrepresented classes.

**iNaturalist2018** The results presented in Tab. 7 demonstrate that our method surpasses several baseline approaches, achieving an overall performance improvement of $1.91\%$ compared to RAC, a method that leverages additional data for training. This performance gain is particularly noteworthy considering the scale of the iNaturalist2018 dataset, which comprises 8,142 distinct classes. The complexity of large-scale classification tasks often introduces significant challenges due to the vast number of classes and the inherent class imbalance. Despite these obstacles, our method proves to be highly effective, outperforming competitors even without the need for extra training data.

## 6.2 ABLATION STUDY

**The influence of incomplete semantic factor** The most critical hyperparameter in our backdoor adjustment method is the number of incomplete semantic factors, denoted as $M$. In our experiments, we set $M = 3$, utilizing the incomplete semantic factors from CLIP, OpenCLIP, and MetaCLIP to obtain the final prediction score. To further investigate the impact of $M$ on performance, we conducted experiments shown in Tab. 8. As the number of incomplete semantic factors increases,

Table 8: The ablation study with the different number of semantic factors of $M$.

| | CLIP | OpenCLIP | MetaCLIP | ImageNet-LT | | | Places365-LT | | |
|---|---|---|---|---|---|---|---|---|---|
| | | | | D-Many | D-Medium | D-Few | D-Many | D-Medium | D-Few |
| $M=1$ | ✔ | | | 80.25 | 76.05 | 71.53 | 51.25 | 52.28 | 50.90 |
| | | ✔ | | 79.94 | 76.17 | 71.70 | 51.71 | 52.20 | 51.77 |
| | | | ✔ | 80.45 | 77.06 | 72.64 | 51.42 | 52.26 | 50.61 |
| $M=2$ | ✔ | ✔ | | 81.65 | 77.99 | 73.91 | 52.28 | 53.31 | 52.31 |
| | ✔ | | ✔ | 81.86 | 78.26 | 74.33 | 52.38 | 53.47 | 52.13 |
| | | ✔ | ✔ | 81.64 | 78.02 | 74.08 | 52.26 | 53.32 | 51.90 |
| $M=3$ | ✔ | ✔ | ✔ | **82.21** | **78.75** | **74.99** | **52.52** | **53.62** | **52.54** |

our backdoor adjustment demonstrates improved performance. For instance, $M = 3$ outperforms $M = 1$, with OpenCLIP showing improvements of **2.27**%, **2.58**%, and **3.29**% on the "D-Many," "D-Medium", and "D-Few" of the ImageNet-LT dataset, respectively. Notably, the most significant improvement was observed in the "D-Few" classes, where the tail classes benefit greatly from a more balanced prediction due to the diverse range of incomplete semantic factors used. This implies that our method is particularly effective at improving the performance on long-tailed distributions, where the tail classes usually suffer from inadequate training samples and biased parameter distributions.

**Backdoor adjustment relieves the parameter imbalance** To verify that our method effectively alleviates parameter imbalance, we report the performance results in Fig. 6, using the division principles that align with the parameter imbalance of CLIP. When compared with zero-shot (ZS) and cross-entropy (CE) baselines in Fig. 2, our backdoor adjustment achieves an overall improvement across all groups. For instance, in the "P-Few" group, our method provides performance gains of **1.32**%, **1.42**%, and **1.50**% over Logit Adjustment (LA) in the "D-Many", "D-Medium", and "D-Few" groups, respectively. These results demonstrate that our method effectively enhances the performance of tail classes while maintaining or improving the performance of head classes, showcasing its ability to balance performance across the board without negative trade-offs.

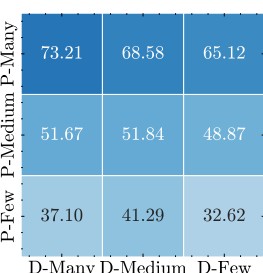

Figure 6: The performance of different groups with our method on Places365-LT.

### 6.3 DISCUSSION

Our method tackles the parameter imbalance in PEFT under long-tailed distributions, providing a balanced solution that improves both model performance and fairness. As shown in Tab. 9, increasing $M$ allows for the identification of more incomplete semantic factors, enhancing the model's ability to address these confounding elements. However, this improvement comes with substantially higher computational costs, underscoring the importance of optimizing $M$ to balance performance gains and efficiency, particularly in resource-constrained scenarios.

Table 9: Computation costs.

| | FLOPs |
|---|---|
| $M=1$ | 239.06 |
| $M=2$ | 478.13 |
| $M=3$ | 717.19 |

## 7 CONCLUSION

In this paper, we analyze how the bias of foundation models influences downstream imbalanced tasks. We formally define parameter imbalance and data imbalance to guide our analysis. After fine-tuning, while data imbalance can be effectively addressed, parameter imbalance persists, hindering further performance improvements. We analyze that the re-balancing based method cannot give many benefits to the feature representation. To address this issue, we solve this problem from another perspective and construct a causal structure graph, identifying the incomplete semantic factors generated by parameter imbalance as key confounders. Consequently, we propose a backdoor adjustment method to mitigate the negative impact of this textcolor blueincomplete semantic factors. Experimental results demonstrate that our method enhances generalization performance across various long-tailed datasets. In future work, we plan to extend our method to other tasks, such as object detection and segmentation under long-tailed distributions.

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

# A   DATASET AND EXPERIMENTAL SETUP

## A.1   DATASET INTRODUCTION

**ImageNet-LT** ImageNet-LT (Liu et al., 2019) is a subset of the ImageNet (Deng et al., 2009) dataset, designed to address the challenges of long-tailed distributions. It contains a total of 12.21K images, with the number of samples per class varying significantly—from 1280 samples for the most represented class to just 5 for the least represented class. The distribution of samples per class is determined by a down-sampled Pareto distribution, emphasizing the disparity in class representation.

**Places365-LT** Places365-LT (Liu et al., 2019) is derived from the Places-2 dataset and consists of 62.5K images spread across 365 categories. This dataset exhibits a more pronounced imbalance compared to ImageNet-LT. In Places365-LT, the largest class contains 4980 images, while the smallest class is represented by only 5 images, resulting in an imbalance factor (IF) of 996.

**iNaturalist2018** The iNaturalist2018 dataset (Van Horn et al., 2018) is focused on natural biological images and presents a significant challenge due to its heavily imbalanced distribution. It encompasses 437.5K images across 8142 classes, making it one of the largest datasets in terms of class diversity. Additionally, iNaturalist2018 poses a fine-grained classification challenge, as it requires distinguishing between similar species and categories, adding another layer of complexity to the classification task.

## A.2   EXPERIMENTAL SETUP

We present the details about the hyper-parameters of our experiments on different datasets in Tab. 10, where lr, epochs denote the initial learning rate and training epochs, respectively. We denote batch_size in Tab. 10 as the training batch size during the fine-tuning phase.

For training resources, all experiments are conducted on Intel(R) Xeon(R) Gold 5318Y CPU @ 2.10GHz with a single RTX A40 GPU. Normally, a GPU with 24GB of memory is sufficient for the reproduction.

Table 10: Hyper-parameters used in our experiments on different datasets.

| Dataset | Lr | Epochs | Batch_size |
|---|---|---|---|
| Places365-LT | 0.01 | 10 | 128 |
| ImageNet-LT | 0.01 | 10 | 128 |
| iNaturalist2018 | 0.01 | 20 | 128 |

## A.3   PARAMETER IMBALANCE AND DATA IMBALANCE

Previous methods have effectively addressed data imbalance by providing a clear definition of the issue. Following the approach of OLTR (Liu et al., 2019), we categorize the classes into three groups: "D-Many", "D-Medium", and "D-Few". Specifically, classes with more than 100 training samples are classified as "D-Many", those with 20–100 samples are categorized as "D-Medium", and classes with fewer than 20 samples fall into the "D-Few" group. This categorization allows for a systematic examination of the impact of data imbalance across varying levels of sample availability.

To address parameter imbalance, where access to pre-training data is unavailable, we adopt the estimation method provided by GLA (Zhu et al., 2024) (Eq. 3) to estimate the label prior. Using this estimation, we divide the classes uniformly into three groups: the top 30% of classes based on the label prior are designated as "P-Many", the next 30–60% as "P-Medium", and the remaining 40% as "P-Few". This approach provides a structured way to assess parameter imbalance across different subsets of classes.

By applying these criteria, any given dataset can be simultaneously split based on both parameter imbalance and data imbalance. Importantly, a single class can belong to multiple groups across the two categorizations; for instance, a class may be grouped as "P-Many" while also being classified

as "D-Few". This dual categorization facilitates a more nuanced analysis of the interplay between parameter and data imbalance.

## B    DETAILED DISCUSSIONS ON INCOMPLETE SEMANTIC FACTORS

We first provide the definition of confounders in the causal framework: When a third variable $Z$ influences both $X$ and $Y$, we say that $X$ and $Y$ are confounded. Such a variable $Z$ is referred to as a confounder of $X$ and $Y$ (Pearl, 2009). Specifically, it satisfies the fork structure $X \leftarrow Z \rightarrow Y$. This latent common cause $Z$ can create a spurious correlation between $X$ and $Y$, making the observed statistical relationship $\mathbb{P}(Y|X)$ potentially misleading. To address this, the causal relationship $\mathbb{P}(Y|\text{do}(X))$ is used as a replacement for the correlation $\mathbb{P}(Y|X)$.

Secondly, we explain why the incomplete semantic factor $C$ in our problem setting can create a backdoor path $X \leftarrow C \rightarrow Y$, leading to a spurious correlation. The example in the first row in Fig.4 explains why the parameter-imbalanced model $D$ leads to an incomplete semantic factor $C$. For example, the semantic factor available to OpenCLIP is in the head of the object, while the semantic factor available to MetaCLIP is in the body of the object.

We emphasize that the semantic factors obtained due to model imbalance are incomplete. Therefore, for rigor, we have modified the causal diagram in Fig. 5 to include the inaccessible semantic factor $U$. From the perspective of data generation, $C$ and $U$ together constitute $X$, thus $X \leftarrow C$ holds. On the other hand, the path $C \rightarrow Y$ indicates that the predicted label distributions follow their own parameter imbalance. A parameter-imbalanced model, OpenCLIP, consistently predicts a "head of the object" label distribution (i.e., making more accurate predictions on test samples where the head of the object is not occluded), regardless of the test distributions. Similarly, the parameter-imbalanced model MetaCLIP exhibits a distinct "body of the object" preference across different test distributions.

The relationship between the confounding path $X \leftarrow C \rightarrow Y$ is unstable. When the training set model predicts the label with the help of the body semantics of the object, if the body of a sample of a certain class in the test example is occluded, the model will give an incorrect prediction. From this example, it can be seen that an ideal model needs to learn the causal relationship $\mathbb{P}(Y|do(X))$ between the input and the label, rather than fitting the unstable confounding path $X \leftarrow C \rightarrow Y$, in order to achieve generalization between different distributions.

## C    INTRODUCTION TO THE VIT AND PEFT

### C.1    VISION TRANSFORMER

The Vision Transformer (ViT) is a deep learning model designed to process image data by leveraging the architecture of transformers, originally developed for natural language processing tasks. Unlike traditional convolutional neural networks (CNNs), which rely on convolutional layers to capture spatial relationships in images, ViT treats an image as a sequence of patches. Each image is divided into fixed-size patches, and these patches are embedded and processed as tokens, similar to words in a sentence. Through self-attention mechanisms, ViT captures global dependencies between patches, allowing it to effectively model long-range relationships within images. ViT has demonstrated strong performance in a variety of computer vision tasks, often surpassing CNNs, particularly as the availability of large-scale image datasets has grown.

For a pre-trained Vision Transformer (ViT) composed of $N$ blocks, denoted as $B = \{B_1, B_2, ..., B_N\}$, each input image $\boldsymbol{x}_i$ is divided into $m$ patches, resulting in patch embeddings $\boldsymbol{E}_i^{(0)} = \{\boldsymbol{e}_{i,1}^{(0)}, \boldsymbol{e}_{i,2}^{(0)}, ..., \boldsymbol{e}_{i,m}^{(0)}\}$. Together with the classification token (CLS), these patch embeddings are passed through the ViT backbone, producing an output embedding $\{\text{CLS}_i^{(N)}, \boldsymbol{e}_{i,1}^{(N)}, \boldsymbol{e}_{i,2}^{(N)}, ..., \boldsymbol{e}_{i,m}^{(N)}\}$.

Each block consists of a multi-head self-attention mechanism followed by a feed-forward layer, both of which incorporate layer normalization and a residual connection. In the self-attention layer, the patch embeddings and the class token are updated based on the similarity matrix, as shown in Eq. 8. Here, $\boldsymbol{Q}_i^{(n-1)}$, $\boldsymbol{K}_i^{(n-1)}$, and $\boldsymbol{V}_i^{(n-1)}$ are derived from $[\text{CLS}_i^{(n-1)}, \boldsymbol{E}_i^{(n-1)}]$ and are input to different

Table 11: The ablation study with the different number of semantic factors of $M$ for VPT.

| | CLIP | OpenCLIP | MetaCLIP | ImageNet-LT | | | Places365-LT | | |
|---|---|---|---|---|---|---|---|---|---|
| | | | | D-Many | D-Medium | D-Few | D-Many | D-Medium | D-Few |
| $M=1$ | ✔ | | | 78.63 | 75.42 | 71.25 | 50.70 | 51.72 | 49.07 |
| | | ✔ | | 78.62 | 75.40 | 70.91 | 51.45 | 51.53 | 50.68 |
| | | | ✔ | 79.86 | 76.47 | 72.19 | 50.91 | 52.06 | 49.86 |
| $M=2$ | ✔ | ✔ | | 80.42 | 77.17 | 73.19 | 51.80 | 52.66 | 51.33 |
| | ✔ | | ✔ | 80.75 | 77.96 | 73.54 | 51.55 | 53.20 | 50.82 |
| | | ✔ | ✔ | 80.54 | 77.52 | 73.41 | 51.89 | 53.09 | 51.22 |
| $M=3$ | ✔ | ✔ | ✔ | **81.09** | **78.37** | **74.16** | **51.97** | **53.26** | **51.34** |

linear transformations within block $B_n$, where $d$ represents the feature dimension. The self-attention layer updates the tokens based on the values in the self-attention matrix.

$$[\widetilde{\mathrm{CLS}}_i^{(n-1)}, \widetilde{\boldsymbol{E}}_i^{(n-1)}] = \mathrm{softmax}(\frac{(\boldsymbol{Q}_i^{(n-1)})^T \boldsymbol{K}_i^{(n-1)}}{\sqrt{d}})\boldsymbol{V}_i^{(n-1)} \tag{8}$$

where $d$ is the feature dimension. The resulting tokens $[\widetilde{\mathrm{CLS}}_i^{(n-1)}, \widetilde{\boldsymbol{E}}_i^{(n-1)}]$ from the multi-head self-attention layer are then passed into the feed-forward layer, producing the output $[\mathrm{CLS}_i^{(n)}, \boldsymbol{E}_i^{(n)}]$ of block $B_n$.

## C.2 ADAPTFORMER

AdaptFormer is a lightweight adaptation framework designed to enhance the efficiency and flexibility of large pre-trained vision transformers (ViTs) in downstream tasks. By introducing a low-rank adaptation (LoRA) module within the feed-forward layers of the transformer, AdaptFormer allows for the modification of the model's capabilities without requiring a full re-training of all parameters. This approach preserves the pre-trained knowledge in the original transformer while enabling it to adapt effectively to new tasks with minimal additional parameters. As a result, AdaptFormer achieves strong performance in transfer learning tasks, providing a balance between computational efficiency and task-specific adaptability.

Let $\boldsymbol{X}$ be the input to the feed-forward layer, and the original weight matrix of the layer be $\boldsymbol{W} \in \mathbb{R}^{d \times d}$. In AdaptFormer, an additional low-rank module is added as a modification. This is represented by two matrices, $\boldsymbol{A} \in \mathbb{R}^{d \times r}$ and $\boldsymbol{B} \in \mathbb{R}^{r \times d}$, where $r \ll d$ represents the low-rank dimension. The output of the modified feed-forward layer is given by:

$$\boldsymbol{Y} = \boldsymbol{W}\boldsymbol{X} + \alpha(\boldsymbol{A}\boldsymbol{B})\boldsymbol{X} \tag{9}$$

Here, $\alpha$ is a scaling factor that controls the influence of the low-rank adaptation. The term $\boldsymbol{A}\boldsymbol{B}$ represents the low-rank adaptation added to the original weight transformation, allowing the model to adapt with a minimal number of additional parameters.

## C.3 VISUAL PROMPT TUNING

Visual Prompt Tuning is an innovative approach designed to adapt pre-trained vision models for specific tasks by utilizing learnable visual prompts. Instead of fine-tuning all the parameters of a large vision model, this method introduces a small set of visual prompts that can be learned during the adaptation process. These prompts are essentially additional visual tokens that are concatenated to the input image embeddings, guiding the model to focus on relevant features for the target task. This approach significantly reduces the computational burden associated with full model fine-tuning while retaining the rich representations learned from extensive pre-training. By leveraging visual prompts, practitioners can achieve competitive performance in various computer vision applications, such as image classification and object detection, with minimal adjustments to the model architecture.

The process of Visual Prompt Tuning can be represented mathematically by introducing a set of learnable visual prompts into the embedding space of a pre-trained vision model. Let $\boldsymbol{P} = $

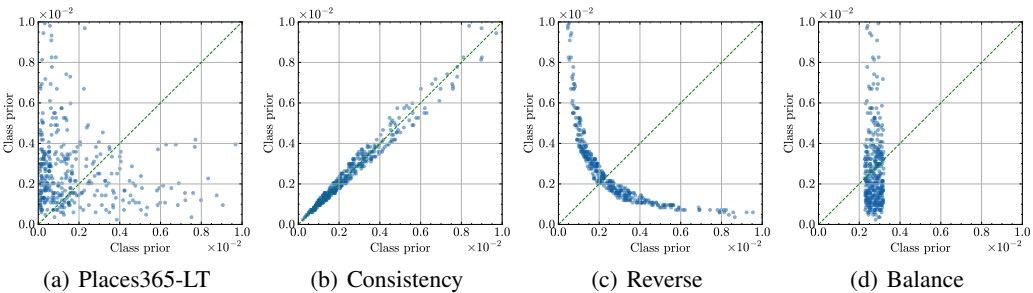

|           | (a) Places365-LT | (b) Consistency | (c) Reverse | (d) Balance |
|-----------|------------------|-----------------|-------------|-------------|

Figure 7: We constitute different parameter imbalanced datasets (a) the original Places365-LT, (b) the data imbalance and parameter imbalance are consistent, (c) the data imbalance and parameter imbalance are reversed, and (d) the downstream dataset is balanced. Each dot in the figure represent the pair $(\mathbb{P}_S(Y=y), \widehat{\mathbb{P}}_P(Y=y))$.

Table 12: The results of decoupling training and LA on ImageNet-LT dataset.

|            | D-Many | D-Medium | D-Few | All   |
|------------|--------|----------|-------|-------|
| LA         | 80.25  | 76.05    | 71.53 | 77.06 |
| Decoupling | 80.30  | 75.90    | 70.4  | 76.80 |

$\{\boldsymbol{P}^{(1)}, \cdots, \boldsymbol{P}^{(N)}\}$, where $\boldsymbol{P}^{(n)} \in \mathbb{R}^{l \times d}$ is the prompts of the $n$-th block., where $l$ is the number of prompts. The visual prompt tuning process can be described by the following equation:

$$\text{CLS}_i^{(n)}, \boldsymbol{E}_i^{(n)} = B_n(\text{CLS}_i^{(n-1)}, \boldsymbol{P}^{(n-1)}, \boldsymbol{E}_i^{(n-1)}) \tag{10}$$

This concatenated representation is then fed into the transformer architecture of the vision model to adapt it for specific downstream tasks. The learnable visual prompts $\boldsymbol{P}$ are updated during the tuning process to optimize task performance while keeping the original model parameters fixed.

## D  ADDITIONAL EXERPIMENTS

### D.1  THE PERFORMANCE FOR VISUAL PROMPT TUNING

We also conduct experiments using different PEFT-based methods, such as Visual Prompt Tuning (VPT), with the results shown in Tab. 11. A similar phenomenon is observed with Adaptorformer for fine-tuning the foundation model. For instance, when $M = 3$ outperforms $M = 1$, OpenCLIP demonstrates improvements of **2.38**%, **2.97**%, and **3.25**% on the "D-Many", "D-Medium", and "D-Few" groups of the ImageNet dataset, respectively. These experiments further indicate that our method generalizes well to different types of PEFT-based methods.

### D.2  THE INFLUENCE OF CLASSIFIERS

Decoupling training (Kang et al., 2019) has demonstrated its effectiveness under long-tailed distributions, suggesting that a generalizable representation can be learned without re-balancing. However, this conclusion is largely based on experiments using CNN architectures trained from scratch, and its influence on fine-tuning-based methods remains unclear. To explore this, we conducted experiments following a similar pipeline, where the backbone is fine-tuned using Cross Entropy (CE) while the classifier is trained with Logit Adjustment (LA). As shown in Tab. 12, decoupling training achieves a performance similar to LA. Combined with previous findings, we observe that LA primarily alleviates data imbalance by driving a more balanced classifier, though it offers limited improvement in the representation of tail classes.

### D.3  THE INFLUENCE OF DATA IMBALANCE ALONE

In our paper, we address both data imbalance and parameter imbalance simultaneously, with analyses primarily based on commonly used datasets. To better understand the impact of parameter

Table 13: The results of Uniform, Resverse, and Balance.

| Balance | Uniform | Reverse |
|---------|---------|---------|
| 53.63 | 50.80 | 51.2 |

Table 14: The imbalanced factor of parameter imbalance of different foundation models on Places365-LT.

| | CLIP | OpenCLIP | MetaCLIP |
|---|------|----------|----------|
| IF | 57.50 | 63.25 | 60.20 |

imbalance, we consider the following scenarios: (1) Consistency: The data imbalance and parameter imbalance are consistent, where the head classes grouped by the data imbalance are also the head classes grouped by the parameter imbalance. (2) Reverse: The data imbalance and parameter imbalance are reversed, where the head classes grouped by the data imbalance are the tail classes grouped by the parameter imbalance. (3)Balance: The downstream data is balanced. To simulate these scenarios, we construct additional imbalanced datasets by downsampling the Places365 dataset, as illustrated in Fig. 7. We evaluate performance using the Places365-LT balanced test set. To reduce the influence of randomness, we create three datasets for each scenario and report the average performance across them.

The results, shown in Tab. 13, indicate that when the downstream data is balanced, the model achieves the best performance, as this setup aligns most closely with the distribution of the downstream task. Interestingly, the Reverse scenario outperforms the Consistency scenario. In the Reverse setting, the tail classes (in terms of parameter imbalance) can be viewed as head classes (in terms of data imbalance), suggesting that parameter imbalance can be mitigated by increasing the number of training samples for tail classes. However, in real-world applications, collecting sufficient data for tail classes is often challenging, and simply adding more data is not a practical solution to this issue.

### D.4 INTREPRETING THE CONFOUNDER

In this section, we conduct experiments to investigate why the incomplete semantic factor acts as a confounder in the causal graph. As illustrated in the first row of Fig. 4, the fine-tuned OpenCLIP model predominantly focuses on the head, whereas MetaCLIP primarily focuses on the body. We define $C = 0$ to represent the head and $C = 1$ to represent the body for this particular class.

To assess the influence of the incomplete semantic factor, we randomly select samples from this class and examine the prediction results of OpenCLIP and MetaCLIP. As shown in Fig. 8, when the dog's head is occluded, OpenCLIP tends to produce lower-confidence predictions. Conversely, when the body is occluded and the head remains visible, OpenCLIP is more likely to provide higher-confidence predictions. Since both models are fine-tuned using the same principle, the observed

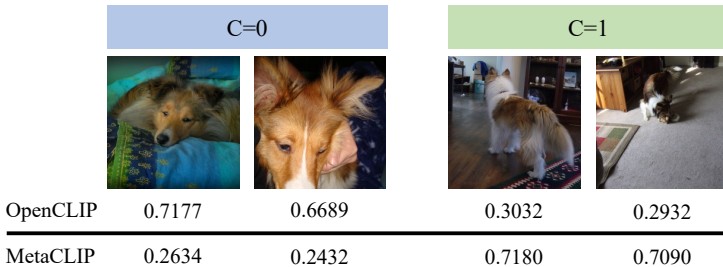

| | C=0 | | C=1 | |
|---|------|------|------|------|
| OpenCLIP | 0.7177 | 0.6689 | 0.3032 | 0.2932 |
| MetaCLIP | 0.2634 | 0.2432 | 0.7180 | 0.7090 |

Figure 8: Prediction scores of fine-tuned OpenCLIP and MetaCLIP. OpenCLIP is more confident about the sample if the head is exposed ($C = 0$), while MetaCLIP is more sensitive to the body ($C = 1$).

Table 15: Differences between $\mathbb{P}(Y \mid X)$ and $\mathbb{P}(Y \mid do(X))$. Image1, Image2, Image3, and Image4 denote the image in Fig. 8 from left to right.

|  | Image1 | Image2 | Image3 | Image4 |
|---|---|---|---|---|
| $\mathbb{P}(Y \mid do(X))$ | 0.5614 | 0.5443 | 0.5837 | 0.5742 |
| $\mathbb{P}(Y \mid X)$ | 0.5431 | 0.2221 | 0.2210 | 0.1031 |

Figure 10: The comparison of our method with other fine-tuning based methods.

differences in predictions can be attributed to incomplete semantic factors, thereby supporting our hypothesis.

To show the existence of confounding bias directly, we measure differences between $\mathbb{P}(Y \mid X)$ and $\mathbb{P}(Y \mid do(X))$. As shown in Tab. 15, there is a significant difference between $\mathbb{P}(Y \mid X)$ and $\mathbb{P}(Y \mid do(X))$, which also verifies the existence of confounding bias.

### D.5 MORE EXPERIMENTSPARAMETER IMBALANCE

In addition, to better support our point, we train a model from scratch on ImageNet-LT with ResNet-50. Since the parameter of ResNet-50 He et al. (2016) is randomly initialized, the trained model is only influenced by the data imbalance. After getting the model, we calculate the accuracy of each class and sort them relying on the estimated pre-training data label prior $\widehat{\mathbb{P}}_P(Y)$. Since the parameter imbalance does not influence the trained model, the performance curve after sorting should not present a downward trend. As shown in Fig. 8, the result verifies our point. Therefore, in this comparison experiment, the downward trend in the right image in Fig. 3 is attributed to the imbalance of pre-training data, which can also verify the impact of parameter imbalance.

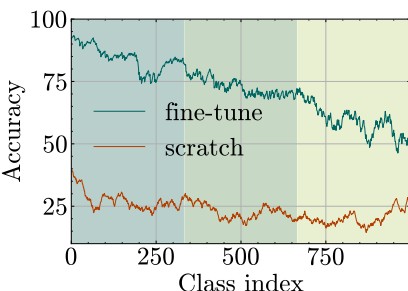

Figure 9: The comparison between fine-tuned model and training from scratch.

### D.6 VISUALIZATION OF OUR BACKDOOR ADJUSTMENT

As illustrated in Fig. 10, we present heatmaps generated after applying our backdoor adjustment, showcasing the effectiveness of our method in mitigating the influence of incomplete semantic factors. By addressing these confounders, our method enhances focus on the entire object rather than isolated parts.

In the first row, OpenCLIP primarily concentrates on the head, while MetaCLIP shifts its attention to the body. In contrast, our method successfully integrates both the head and body, yielding a more holistic understanding of the object. Similarly, in the second row, our method expands its

Table 16: The six classes and their corresponding incomplete semantic factors.

| | | |
|---|---|---|
| airplane | airplane head | airplane tail |
| automobile | automobile head | automobile tail |
| bird | bird head | bird body |
| cat | cat head | cat body |
| dog | dog head | dog body |
| truck | truck head | truck tail |

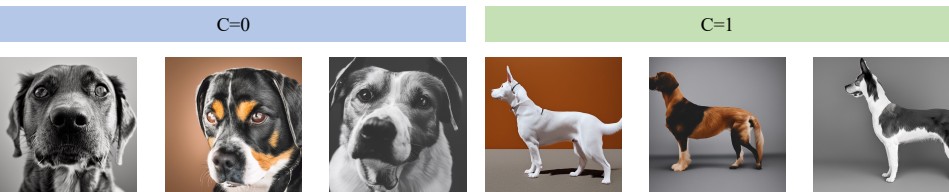

Figure 11: The visualization of the "dog" class: $C = 0$ represents samples generated for the dog's head, while $C = 1$ corresponds to samples of the dog's body.

focus to include the head, wings, and tail, offering a more comprehensive representation of the object. Finally, in the last row, our method effectively captures the entire structure of the truck, demonstrating its robustness and adaptability to diverse objects and scenarios.

### D.7 EXPERIMENTAL EVIDENCE FOR $C \to Y$

In this section, we will verify the existence of $C \to Y$ in experiments. We constructed a dataset consisting of six classes: "airplane", "automobile", "bird", "cat", "dog", and "truck". For each class, we define distinct incomplete semantic factors, as shown in Tab. 16, where $C$ denotes the incomplete semantic factor:

For example, the term "airplane head" refers to the front part of the aircraft, including the cockpit, while "airplane tail" represents the rear section, including the winglets. To construct the dataset, we used Stable Diffusion (Rombach et al., 2022) by providing prompts in the format: "a photo of a {class name}'s {head or tail}". For instance, to generate samples for the airplane class, we used the prompts "a photo of an airplane's head" and "a photo of an airplane's tail", respectively, ensuring sufficient samples for each semantic factor. We also visualize some samples in Fig. 11.

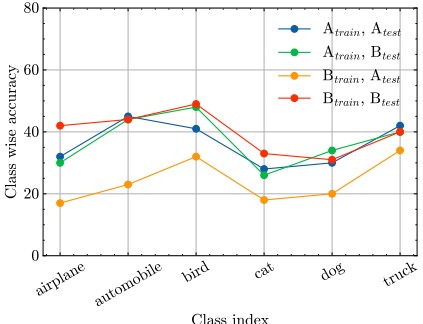

Figure 12: The performance curve illustrates four evaluation scenarios: $A_{train}, A_{test}$ indicates the model is trained on $A_{train}$ and evaluated on $A_{test}$; $A_{train}, B_{test}$ represents the model trained on $A_{train}$ and evaluated on $B_{test}$; $B_{train}, A_{test}$ corresponds to the model trained on $B_{train}$ and evaluated on $A_{test}$; and $B_{train}, B_{test}$ denotes the model trained on $B_{train}$ and evaluated on $B_{test}$.

The training dataset is created under the following conditions: 1) For each incomplete semantic factor, 50 samples are generated. This dataset is referred to as $A_{train}$. 2) For $C = 0$, 90 samples are generated, and for $C = 1$, 10 samples are generated. This dataset is referred to as $B_{train}$.

For evaluation, the test dataset is constructed under the following conditions: 1) For each incomplete semantic factor, 50 samples are generated. This dataset is referred to as $A_{test}$. 2) For $C = 0$, 90 samples are generated, and for $C = 1$, 10 samples are generated. This dataset is referred to as $B_{test}$.

We use ResNet-32 (He et al., 2016) as the backbone and train it from scratch on $A_{train}$ and $B_{train}$, respectively. The trained models are then evaluated on $A_{test}$ and $B_{test}$, respectively. As shown in Fig. 12, when the model is trained on $B_{train}$ but evaluated on $A_{test}$, we find the performance drops (compared with testing on $B_{test}$), which indicates the confounder influences the model and cannot make right predictions. Our findings can be summarized as follows:

1. The model trained on $A_{train}$, when used for inference, can be interpreted as estimating $\mathbb{P}(Y \mid do(X))$. We observe that the model trained on $A_{train}$ performs similarly on both $A_{test}$ and $B_{test}$. This result indicates that the trained model does not learn the correlation.

2. The model trained on $B_{train}$, when used for inference, can be interpreted as estimating $\mathbb{P}(Y \mid X)$. We observe that the model trained on $B_{train}$ exhibits different performance between $A_{test}$ and $B_{test}$. This difference arises due to the influence of the confounder.

From these experiments, we verify that $C$ is a confounder.