# OpenReview forum: "Rethinking the Bias of Foundation Model under Long-tailed Distribution"
_ICLR.cc/2025/Conference — Submitted to ICLR 2025_

### Official Review · Reviewer_MpPT · 2024-10-27

**Soundness:** 2
**Presentation:** 1
**Contribution:** 2
**Rating:** 6
**Confidence:** 2

**Summary:**

The paper aims to solve the problem of long-tailed learning and illustrates the limitations of rebalancing methods due to biases in foundation models resulting from the imbalance in pre-training data. Based on a causal structure graph and back-door adjustment, the paper proposes a method to mitigate the spurious correlations introduced to the foundation model.

**Strengths:**

1. The paper identifies the importance of parameter imbalance and thoroughly analyzes the phenomenon with examples. It also highlights the limitations of logic adjustment-based methods in mitigating parameter imbalance.

2. The authors put significant effort into conducting thorough experiments.

**Weaknesses:**

1. The writing can be greatly improved. For example, the paper mentions Figures 2 and 3 in the introduction without further explaining what they mean. More explanation on figure 2 and 3 in the introduction or detailed caption for figures 2 and 3 would be preferred. Authors also refer to “D-many,” “D-Medium,” and “D-Few” without explicitly explaining what they are. Given the frequency of these terms, I would expect more details on what they means e.g., the definition and data processing etc.

2. Regarding the experimental results,  the improvement of the proposed method looks slightly marginal given that fine-tuning based method can also alleviate the bias. Maybe a side-by-side comparison of the improvement of the proposed method and fine-tuning based method would be beneficial. Moreoever, the results are mostly in tables, visual examples similar to those in Figure 4, that illustrate how the method reduces spurious correlations could be beneficial. To be more precise, what does the heatmap looks like after the application of the proposed method? Has the problem of partial semantic factor been alleviated from the visualization?

**Questions:**

1. Could you explain a bit more about the concept of partial semantic factor and the causal graph in Section 5.1? What is a data-balanced representation, and could you provide an example of  $C \rightarrow X$?

2. In Section 4.1, the authors claim that although the fine-tuning-based method is able to address parameter imbalance, bias still exists; the results shown in figure 3 for Places365 looks unreasonably bad and could you provide more details on this part of experiments?

---

> ### Author Response · Authors · 2024-11-20
>
> We greatly appreciate your detailed feedback and thoughtful suggestions. To ensure all your concerns are thoroughly addressed, we have summarized your comments in quotes below and provided our responses to each point in detail.
>
> **Weakness 1**
> >The writing can be greatly improved. For example, the paper mentions Figures 2 and 3 in the introduction without further explaining what they mean. More explanation on figure 2 and 3 in the introduction or detailed caption for figures 2 and 3 would be preferred. Authors also refer to “D-many,” “D-Medium,” and “D-Few” without explicitly explaining what they are. Given the frequency of these terms, I would expect more details on what they means e.g., the definition and data processing etc.
>
> **Re:**
> In the introduction, we provide a brief summary of our findings and approach. A more detailed explanation of these two figures is available in Section 4.1. To clarify, we have added further details regarding these images in the captions in the revised version.
>
> Regarding data imbalance, we follow the approach outlined in OLTR [a] and categorize the classes into "D-Many," "D-Medium," and "D-Few." Specifically, we group classes with more than 100 training samples as "D-Many," those with 20-100 samples as "D-Medium," and those with fewer than 20 samples as "D-Few."
>
> For parameter imbalance, given the unavailability of pre-training data, we use GLA [b] to estimate the label prior and uniformly split the classes. We designate the top 30% of classes as "P-Many," the next 30%-60% as "P-Medium," and the remaining classes as "P-Few." For example, if class $A$ in the downstream task has 1000 samples, it would be grouped as "D-Many." However, if class $A$ has a lower estimated pre-training prior, it would be also classified as "P-Few". Therefore, its performance in Fig.2 should be grouped into "D-Many" (column 1) and "P-Few" (row 3).
>
> To further clarify these concepts, we have added additional details in the appendix.
>
> **Weakness 2**
> >Regarding the experimental results, the improvement of the proposed method looks slightly marginal given that fine-tuning based method can also alleviate the bias. Maybe a side-by-side comparison of the improvement of the proposed method and fine-tuning based method would be beneficial. Moreoever, the results are mostly in tables, visual examples similar to those in Figure 4, that illustrate how the method reduces spurious correlations could be beneficial. To be more precise, what does the heatmap looks like after the application of the proposed method? Has the problem of partial semantic factor been alleviated from the visualization?
>
> **Re:**
> Thank you for your valuable feedback. Indeed, the results for  $M=1$ in Tab.8 can be interpreted as a fine-tuning-based approach. As $M$ increases, the performance gradually improves, highlighting the effectiveness of our method.
>
> We have also visualized the heatmap of our method when $M=3$. As shown in Fig. 10 in the paper, in the first row, OpenCLIP primarily concentrates on the head, while MetaCLIP shifts its attention to the body. In contrast, our method successfully integrates both the head and body, yielding a more holistic understanding of the object. Similarly, in the second row, our method expands its focus to include the head, wings, and tail, offering a more comprehensive representation of the object. Finally, in the last row, our method effectively captures the entire structure of the truck, demonstrating its robustness and adaptability to diverse objects and scenarios. Our method is able to better focus on the object by eliminating the influence of the accessible semantic factors. We have included these results in the revised version of the paper.

---

> ### Author Response · Authors · 2024-11-20
>
> **Question 1**
> >Could you explain a bit more about the concept of partial semantic factor and the causal graph in Section 5.1? What is a data-balanced representation, and could you provide an example of  $C \rightarrow X$
>
> **Re:**
> Thank you for your valuable feedback. In response, we have reconstructed the causal graph and introduced $C$ and $U$ as **incomplete semantic factors** and inaccessible semantic factors. The path $U \rightarrow X \leftarrow C$ illustrates that the dataset can be generated by considering both incomplete semantic factors and inaccessible semantic factors.
>
> As shown in the first row of Fig. 4, CLIP is more influenced by the bird's wing, so $C$ represents the bird's wing in this case. However, we cannot fully generate the bird by relying solely on the wing (i.e., the incomplete semantic factor). We also need other inaccessible semantic factors, such as the bird's head or tail, which we represent using the variable $U$,  Therefore $U \rightarrow X \leftarrow C$ indicates that the bird's wing, along with other parts of the bird, contributes to generating the bird object.
>
>
> The concept of data balanced representation refers to the learned representation being unaffected by the bias introduced by imbalanced downstream data. When a model is trained without incorporating any rebalancing techniques, the learned representation is dominated by head classes, which leads to the underrepresentation of tail classes. Consequently, we denote $B$ as the balanced representation learned from imbalanced downstream data through data rebalancing methods, such as LA.
>
> **Question 2**
> >In Section 4.1, the authors claim that although the fine-tuning-based method is able to address parameter imbalance, bias still exists; the results shown in figure 3 for Places365 looks unreasonably bad and could you provide more details on this part of experiments?
>
> **Re:**
> Thank you for pointing this out. We conduct experiments on Places365-LT and ImageNet-LT using the CE and LA loss functions. To better investigate the influence of data imbalance and parameter imbalance, we sort the class indices based on the label prior. The x-axis in the left image of Fig. 3 represents the class indices sorted by $P_S(Y)$, while the right image shows the indices sorted by the estimated upstream label prior $\widehat{P}_P(Y)$. Since the pre-training data is inaccessible, we use the method from [b] to estimate the upstream label prior $\widehat{P}_P(Y)$.
>
> As shown in Fig. 3, the performance of tail classes (sorted by $\widehat{P}_P(Y)$) decreases, indicating the presence of parameter imbalance. After applying the LA loss, we observe that performance also decreases, but the rate of decrease weakens. Therefore, we conclude that "fine-tuning-based methods can address parameter imbalance, but bias still exists."
>
> Regarding the performance on Places365-LT, this dataset is inherently more challenging. Typically, we achieve about 79% overall accuracy on ImageNet-LT, but only around 50% on Places365-LT. Thus, when compared to ImageNet-LT, the performance on Places365-LT appears disproportionately poor. However, the absolute lower performance on Places365-LT does not alter our conclusion. We also observe that the LA curve lies above the zero-shot baseline (indicating that parameter imbalance is partly relieved) but still shows a decreasing trend, suggesting that bias remains.
>
>
> [a] Liu, Ziwei, et al. "Large-scale long-tailed recognition in an open world." Proceedings of the IEEE/CVF conference on computer vision and pattern recognition. 2019.
> [b] Beier Zhu, Kaihua Tang, Qianru Sun, and Hanwang Zhang. Generalized logit adjustment: Calibrating fine-tuned models by removing label bias in foundation models. Advances in Neural Information Processing Systems, 36, 2024.

---

> > ### Author Response · Authors · 2024-11-25
> >
> > Dear Reviewer,
> >
> > We sincerely thank you once again for your professional and constructive feedback. We deeply appreciate your suggestion, which has guided us in providing a detailed explanation of our experimental setup and including heatmap visualizations. We believe these additions significantly enhance the depth and clarity of our ablation experiments.
> >
> > As the discussion period is coming to a close, we would be grateful for your feedback on whether our responses and the revised manuscript have sufficiently addressed your concerns or if you have any additional comments to share.
> >
> > Best regards,
> > The Authors

---

> > > ### Comment · Reviewer_MpPT · 2024-11-25
> > >
> > > Thank you for the thorough responses, which have addressed my concerns. But after considering feedback from other reviewers along with the authors' replies, I believe the paper's presentation still needs improvement, particularly in conveying the methodology, the significance of the work, and the experimental details and results. Therefore, I will maintain my current score.

---

> > > > ### Author Response · Authors · 2024-11-25
> > > >
> > > > Thank you for your response. We firmly believe that our work makes a significant contribution to the field of long-tailed learning. Unlike previous studies, our research is the first to explicitly address the issue of pre-training bias in the context of long-tailed distributions. The main contributions of our study are as follows:
> > > >
> > > > 1. We tackle a practical challenge where both the data used to train the foundation model and the data used for adaptation to downstream tasks are imbalanced.
> > > >
> > > > 2. We conduct comprehensive experiments to analyze the impact of pre-training bias under long-tailed distributions, a topic that has not been explored before.
> > > >
> > > > 3. We identify that incomplete semantic factors lead the model to learn spurious correlations between input samples and labels, thereby hindering its generalization ability. To address this, we construct a causal graph and propose a backdoor adjustment method to mitigate the adverse effects of confounders.
> > > >
> > > > In addition, we have revised our paper based on the comments from all the reviewers, further enhancing the quality and clarity of our work.

---

> > > > > ### Author Response · Authors · 2024-11-27
> > > > >
> > > > > Thank you for your time and effort. We have reiterated the methodology, the significance of our work, and the experimental details and results. Following your valuable suggestions, we have carefully revised our paper. As the deadline for uploading the revised PDF is approaching, could you kindly let us know if there are any areas where our paper could be further improved? If you have any questions or concerns, we would be happy to provide detailed responses and make additional revisions based on your feedback.

---

> > > > > > ### Author Response · Authors · 2024-12-02
> > > > > >
> > > > > > Dear Reviewer MpPT,
> > > > > >
> > > > > > We sincerely thank you for your time and thoughtful feedback on our work. Your insights have been invaluable in helping us refine our submission.
> > > > > >
> > > > > > At this point, several reviewers have expressed positive evaluations of the paper. Following substantial revisions and discussions, we believe the manuscript has been significantly improved, with many concerns thoroughly addressed. In light of these updates, we kindly hope you might reconsider your assessment.
> > > > > >
> > > > > > If there are any remaining questions or concerns, we would be more than willing to provide additional clarifications or make further adjustments based on your suggestions.
> > > > > >
> > > > > > Thank you once again for your careful review and constructive feedback.
> > > > > >
> > > > > > Best regards,
> > > > > >
> > > > > > The Authors

---

### Official Review · Reviewer_DbkM · 2024-11-03

**Soundness:** 2
**Presentation:** 3
**Contribution:** 2
**Rating:** 6
**Confidence:** 3

**Summary:**

This work considers the bias of foundation models when the data for fine-tuning suffers from class imbalance. They show that parameter imbalance is more dominant than data imbalance. Then, they propose a method based on an assumed causal graph and backdoor adjustment. Experiments show that the proposed method is effective.

**Strengths:**

- They made a clear observation from empirical results that parameter imbalance is more dominant than data imbalance.

- They show that previous methods such as GLA and re-balancing cannot resolve the parameter imbalance issue.

- The paper is well-written and easy to understand.

**Weaknesses:**

- The argument that C is a confounder is confusing. The model will predict Y using the representation B, it is hard to believe there is a direct causal effect of C on Y. I would suggest the authors to show the direct causal effect or confounding bias with experiments.

- The proposed method utilized \textit{backdoor adjustment}, which is also used in some previous work in the literature of long tail classification such as [1]. The authors did not point out the difference.

[1] Tang, Kaihua, Jianqiang Huang, and Hanwang Zhang. "Long-tailed classification by keeping the good and removing the bad momentum causal effect." _Advances in neural information processing systems_ 33 (2020): 1513-1524.

**Questions:**

1. In Table 8, it seems that as M increases, performance improves. Do the authors think this trend can persist for larger values of M? Is there any theory or insight that can support this observed trend?

2. In Table 7, why can the proposed method also improve performance in D-Many? I suppose such methods would sacrifice the performance on D-Many for improvements on long tail classes.

---

> ### Author Response · Authors · 2024-11-20
>
> Thank you for your insightful feedback and constructive input. We have carefully reviewed your concerns and questions and provided detailed responses to each. Below, we summarize your points in quotes, followed by our corresponding replies.
>
> **Weakness 1**
> >The argument that C is a confounder is confusing. The model will predict Y using the representation B, it is hard to believe there is a direct causal effect of C on Y. I would suggest the authors to show the direct causal effect or confounding bias with experiments.
>
> **Re:**
>
> ***(1)Experimental explanation of why the incomplete semantic factors obtained from a parameter-imbalanced model are confounders.*** Thanks for your advice. We conduct experiments to investigate why the incomplete semantic factor acts as a confounder in the causal graph. As illustrated in the first row of Fig.4, the fine-tuned OpenCLIP model predominantly focuses on the head, whereas MetaCLIP primarily focuses on the body. Accordingly, we define $C=0$ to represent the head and $C=1$ to represent the body for this particular class.
>
> To assess the influence of the incomplete semantic factor, we randomly select samples from this class and examine the prediction results of OpenCLIP and MetaCLIP. As shown in Fig.10, when the dog's head is occluded, OpenCLIP tends to produce lower-confidence predictions. Conversely, when the body is occluded and the head remains visible, OpenCLIP is more likely to provide higher-confidence predictions. Since both models are fine-tuned using the same principle, the observed differences in predictions can be attributed to incomplete semantic factors, thereby supporting our hypothesis.
>
> ***(2)Theoretical explanation of why the incomplete semantic factors obtained from a parameter-imbalanced model are confounders.*** We first provide the definition of confounders in the causal framework: When a third variable $Z$ influences both $X$ and $Y$, we say that $X$ and $Y$ are confounded. Such a variable $Z$ is referred to as a confounder of $X$ and $Y$ [Pearl et al., 2009, p. 55, Def. 2.7.1 & p. 184, Sec. 6.22]. Specifically, it satisfies the fork structure $X \leftarrow Z \rightarrow Y$. This latent common cause $Z$ can create a spurious correlation between $X$ and $Y$, making the observed statistical relationship $P(Y|X)$ potentially misleading. To address this, the causal relationship $P(Y|\text{do}(X))$ is used as a replacement for the correlation $P(Y|X)$.
>
> Secondly, we explain why the incomplete semantic factor $C$ in our problem setting can create a backdoor path $X \leftarrow C \rightarrow Y$, leading to a spurious correlation. The example in the first row in Fig.4 explains why the parameter-imbalanced model $D$ leads to an **incomplete semantic factor** $C$. For example, the semantic factor available to OpenCLIP is in the head of the object, while the semantic factor available to MetaCLIP is in the body of the object.
>
> We emphasize that the semantic factors obtained due to model imbalance are incomplete. Therefore, for rigor, we have modified the causal diagram in Fig. 5 to include the inaccessible semantic factor $U$. From the perspective of data generation, $C$ and $U$ together constitute $X$, thus $X \leftarrow C$ holds. On the other hand, the path $C \rightarrow Y$ indicates that the predicted label distributions follow their own parameter imbalance. A parameter-imbalanced model, OpenCLIP, consistently predicts a "head of the object" label distribution (i.e., making more accurate predictions on test samples where the head of the object is not occluded), regardless of the test distributions. Similarly, the parameter-imbalanced model MetaCLIP exhibits a distinct "body of the object" preference across different test distributions.
>
> The relationship between the confounding path $X \leftarrow C \rightarrow Y$ is unstable. When the training set model predicts the label with the help of the body semantics of the object, if the body of a sample of a certain class in the test example is occluded, the model will give an incorrect prediction. From this example, it can be seen that an ideal model needs to learn the causal relationship $P(Y|do(X))$ between the input and the label, rather than fitting the unstable confounding path $X \leftarrow C \rightarrow Y$, in order to achieve generalization between different distributions.
>
> [Pearl et al., 2009] Pearl, Judea. *Causality*. Cambridge university press, 2009.

---

> ### Author Response · Authors · 2024-11-20
>
> **Weakness 2**
> >The proposed method utilized \textit{backdoor adjustment}, which is also used in some previous work in the literature of long tail classification such as [1]. The authors did not point out the difference.
>
> **Re:**
> In causal inference, the backdoor adjustment is a crucial tool for identifying and controlling confounding variables to enable causal estimation. It primarily addresses the challenge of estimating causal effects in observational data by adjusting for specific variables.
>
> In [1], the authors analyze the SGD momentum as a confounder and apply the backdoor adjustment to mitigate its influence. In our method, we treat the **incomplete semantic factor** as the confounder and disentangle its impact.  The distinctions are as follows:
>
> **(1)Motivation:** Our method investigates how the bias of foundation models influences imbalanced downstream tasks. By contrast, in [1], the authors examine how the momentum in the SGD optimizer affects long-tailed learning.
>
> **(2)Confounder:** We consider the incomplete semantic factor as the confounder, while in [1], the authors identify momentum as the confounder. Consequently, the causal graphs in these two cases are distinct.
>
> **(3)Method:** We implement the computation of $P(Y|do(X))$ through diverse scenarios and methods. In [1], the authors consider the imbalance can lead to momentum bias, and calculate $P(Y|x,m)$ through the de-confounded training to relieve it. In contrast, our method focuses on incomplete semantic factors, computing $P(Y|b,c)$ by leveraging different foundation models to extract varied semantic information, thereby ensuring the model's inference is not constrained to incomplete semantic information.
>
> **Question 1**
> >In Table 8, it seems that as M increases, performance improves. Do the authors think this trend can persist for larger values of M? Is there any theory or insight that can support this observed trend?
>
> **Re:**
> Theoretically, we analyze the problem from the perspective of the Completeness of the Available Value Set for the **Adjustment Variable**. Here, our adjustment variable is $C$. Technically, the possible values of $C$ are infinite and for backdoor adjustment, the sum should account for every possible domain of $C$. A fine-grained definition of $C$ would lead to enormous computational costs, which makes it impractical for training.
> When pre-training datasets correspond to diverse semantic information, an increase in the number of available semantic factors (i.e., a higher $M$) generally enhances performance [a]. However, when $M$ becomes sufficiently large, indicating that potential confounders are more comprehensively addressed, further increasing $M$ yields slight performance improvements. Additionally, a larger $M$ incurs higher computational costs. Therefore, determining the optimal value of $M$ requires balancing performance gains with computational efficiency.
>
> **Question 2**
> >In Table 7, why can the proposed method also improve performance in D-Many? I suppose such methods would sacrifice the performance on D-Many for improvements on long tail classes.
>
> **Re:**
> Thank you for highlighting this point. Improving performance in tail classes does not necessarily come at the expense of performance in head classes. For instance, RIDE [b] demonstrates overall performance improvements across the "Many," "Medium," and "Few" classes compared to other rebalancing-based methods on the iNaturalist2018 dataset.
>
> Additionally, our method emphasizes identifying all available semantic factors across different classes. The "D-Many" also faces a spurious relationship due to incomplete semantic factors that do not depend on downstream data imbalance. Therefore, our method not only supports the classification of tail classes but also enhances the performance of head classes. As a result, our method achieves an overall improvement in classification performance.
>
>
> [a] Li, Jiangmeng, et al. "Hierarchical Topology Isomorphism Expertise Embedded Graph Contrastive Learning." Proceedings of the AAAI Conference on Artificial Intelligence. Vol. 38. No. 12. 2024.
> [b] Wang, Xudong, et al. "Long-tailed recognition by routing diverse distribution-aware experts." arXiv preprint arXiv:2010.01809 (2020).

---

> ### Comment · Reviewer_DbkM · 2024-11-24
>
> > (1)Experimental explanation of why the incomplete semantic factors obtained from a parameter-imbalanced model are confounders.
>
> Thanks for the reply.
>
> The experiment only illustrates the correlation between $C$ and confidence scores. To show the existence of confounding bias, by definition, the authors need to show concretely that $P(Y|do(X)) \ne P(Y|X)$.

---

> ### Author Response · Authors · 2024-11-25
>
> Thank you for your reply. Our backdoor adjustment aims to estimate the probability $P(Y \mid do(X))$, and we present the estimated values as follows. Additionally, to estimate $P(Y \mid X)$, we follow the previous work [a] and train the model from scratch to directly estimate the probability. The results are presented below. For clarity, we refer to the images in Fig. 8 from left to right as Fig. 8(1), Fig. 8(2), Fig. 8(3), and Fig. 8(4).
>
> ||Fig.8(1)|Fig.8(2)|Fig.8(3)|Fig.8(4)|
> |--|--|--|--|--|
> |$P(Y\|do(X))$| 0.5614 |0.5443 | 0.5837 | 0.5742|
> |$P(Y\|X)$| 0.5431 | 0.2221 | 0.2210 |0.1031 |
>
> From these results, we find that $P(Y \mid do(X)) \ne P(Y \mid X) $ and confirm the existence of confounding bias.
>
> If you have any other questions, please feel free to let me know, and I would be happy to respond.
>
> [a] Liu, Ziwei, et al. "Large-scale long-tailed recognition in an open world." Proceedings of the IEEE/CVF conference on computer vision and pattern recognition. 2019.

---

> > ### Comment · Reviewer_DbkM · 2024-11-25
> >
> > I mean the ground truth $P(Y|do(X))$ and $P(Y|X)$, not the estimated ones shown in Fig. 8 because the estimated ones can be arbitrarily biased themselves with no ground truth to verify, especially when $\hat{P}(Y|do(X))$ is estimated under the assumption that $C$ is confounder.
> >
> > I also understand in the authors' experimental setting, it is impossible to know the ground truth. So, I wonder if you can do synthetic experiments where you surely know $C$ is a confounder. Alternatively, the authors can show that $C$ has a direct effect on $Y$ in real-world datasets.

---

> ### Author Response · Authors · 2024-11-25
>
> Thank you for your reply. We constructed a dataset consisting of six classes: 'airplane,' 'automobile,' 'bird,' 'cat,' 'dog,' and 'truck.' For each class, we define distinct incomplete semantic factors, as shown below, where $C$ denotes the incomplete semantic factor:
>
> ||$C=0$|$C=1$|
> |--|--|--|
> |airplane|airplane head|airplane tail|
> |automobile|automobile head|automobile tail|
> |bird|bird head |bird body|
> |cat |cat head| cat body|
> |dog|dog head|dog body|
> |truck|truck head| truck tail|
>
> For example, the term "airplane head" refers to the front part of the aircraft, including the cockpit, while "airplane tail" represents the rear section, including the winglets. To construct the dataset, we used Stable Diffusion [a] by providing prompts in the format: "a photo of a {class name}'s {head or tail}." For instance, to generate samples for the airplane class, we used the prompts "a photo of an airplane's head" and "a photo of an airplane's tail," respectively, ensuring sufficient samples for each semantic factor.
>
> The training dataset was created under the following conditions:
>
> 1) For each incomplete semantic factor, 50 samples were generated. This dataset is referred to as $A_{train}$.
>
> 2) For $C=0$, 90 samples were generated, and for $C=1$, 10 samples were generated. This dataset is referred to as $B_{train}$.
>
> For evaluation, the test dataset was constructed under the following conditions:
>
> 1) For each incomplete semantic factor, 50 samples were generated. This dataset is referred to as $A_{test}$.
>
> 2) For $C=0$, 90 samples were generated, and for $C=1$, 10 samples were generated. This dataset is referred to as $B_{test}$.
>
> We use ResNet-32 [b] as the backbone and train it from scratch on $A_{{train}}$ and $B_{{train}}$, respectively. The trained models are then evaluated on $A_{\text{test}}$ and $B_{\text{test}}$, respectively. As shown in Fig.12, when the model is trained on $B_{train}$ but evaluated on $A_{test}$, we find the performance drops (compared with testing on $B_{test}$), which indicates the confounder influences the model and cannot make right predictions. Our findings can be summarized as follows:
>
> 1. The model trained on $A_{{train}}$, when used for inference, can be interpreted as estimating $P(Y \mid {do}(X))$. We observe that the model trained on $A_{{train}}$ performs similarly on both $A_{{test}}$ and $B_{{test}}$. This result indicates that the trained model does not learn the correlation.
>
> 2. The model trained on $B_{{train}}$, when used for inference, can be interpreted as estimating $P(Y \mid X)$. We observe that the model trained on $B_{{train}}$ exhibits different performance between $A_{{test}}$ and $B_{{test}}$. Considering that if the trained model is not affected by incomplete semantic factors, it should present the same performance between $A_{{test}}$ and $B_{{test}}$. However, we find the evaluation results are different, verifying the confounder's influence.
>
>
> From these experiments, we verify that $C$ is a confounder. More details are in the revised paper.
>
> [a]Rombach, Robin, et al. "High-resolution image synthesis with latent diffusion models." Proceedings of the IEEE/CVF conference on computer vision and pattern recognition. 2022.
>
> [b]He, Kaiming, et al. "Deep residual learning for image recognition." Proceedings of the IEEE conference on computer vision and pattern recognition. 2016.

---

> > ### Author Response · Authors · 2024-11-27
> >
> > Thank you for your valuable suggestions. We have carefully revised our paper based on your feedback. As the discussion period has been extended and the deadline for uploading the revised PDF is approaching, please feel free to reach out if you have any further questions or concerns. We would be happy to provide detailed responses and make additional revisions if needed.

---

> ### Comment · Reviewer_DbkM · 2024-11-27
>
> > The model trained on $A_{train}$, when used for inference, can be interpreted as estimating $P(Y|do(X))$.
>
> This does not sound correct to me. In $A_{train}$ you do have P(C=1)=P(C=0), however, there is no intervention on $X$, that means $X$ is still causally affected by $C$ in $A_{train}$. The correct way to compute $P(Y|do(X))$ is to maintain $C$ and intervene on $X$.

---

> > ### Author Response · Authors · 2024-11-28
> >
> > Thanks for your reply. To explain why the model trained on the dataset with $P(C=1)=P(C=0)$ can be interpreted as estimating $P(Y|do(X))$, we would like to use the example of Simpson's paradox [a, b].
> >
> > **(1) Simpson's paradox**
> >
> > In a study investigating the effectiveness of a treatment (denoted as $X$) on recovery rates (denoted as $Y$), the data is split by gender (where gender $C$ is the confounder). Within each gender group (male and female), the treatment appears to have a positive effect: for males, those who receive the treatment recover at a higher rate than those who do not, and similarly for females. However, when the data is combined across both genders, the treatment appears to be less effective overall, with the recovery rate for those who received the treatment being lower than for those who did not. This reversal of the trend is an example of Simpson's paradox, where gender $C$ is a confounder, distorting the true causal relationship between treatment ($X$) and recovery ($Y$).
> >
> > To properly estimate the causal effect of the treatment while controlling for gender as a confounder, we can use the $do(X=x)$ operation. The do operation involves setting the treatment $X$ to a specific value (e.g., administering the treatment to all patients) while removing the influence of gender on both the treatment and recovery. This can be approximated by balancing the confounder $C$ (gender) across the treatment groups. Specifically, **we can ensure that $P(C=1)=P(C=0)$, meaning that males and females are equally represented in both the treated and untreated groups**. By balancing gender in this way, we effectively control for its confounding effect, enabling us to more accurately estimate the causal effect of the treatment on recovery, as if we had intervened on the treatment while controlling for gender.
> >
> > This process is also described in the book [b] on page 57: **"randomization gives us the quantity we seek to estimate, namely $P(Y = y|do(X = x))$."**
> >
> > **(2) Interpreting our experiment**
> > In our experiments, $X$ is the dataset, $C$ is the incomplete semantic factor, and $Y$ is the predicted label. Applying the example of Simpson's paradox above, we constitute two dataset $A_{train}$ ($P(C=1)=P(C=0)$), and $B_{train}$ ($P(C=1)\neq P(C=0)$). The model trained on the dataset with $P(C=1)=P(C=0)$ can be interpreted as estimating $P(Y|do(X))$.
> >
> > For more details about Simpson's paradox, please refer to https://en.wikipedia.org/wiki/Simpson%27s_paradox.
> >
> > Thanks for your feedback again. If you have any further questions or concerns. We would be happy to provide detailed responses and make additional revisions if needed.
> >
> > [a] Wagner, Clifford H. "Simpson's paradox in real life." The American Statistician 36.1 (1982): 46-48.
> >
> > [b] Pearl, Judea. *Causality*. Cambridge university press, 2009.

---

> ### Comment · Reviewer_DbkM · 2024-11-28
>
> Thanks for the reply.
>
> Note that, in your experiments, the datasets $A_{train}$ and $B_{train}$ are generated from the same causal graph/data-generating process. The difference is only the number of samples for $P(C=1)$ and $P(C=0)$. This should not affect the identification of the interventional distribution $P(Y|do(X))$, which is still unclear to me (I would assume that the authors rely on backdoor criteria to estimate $P(Y|do(X)) = \sum_c P(Y|X,C=c)P(C=c)$. In both balanced and imbalanced cases, we should be able to estimate both $P(Y|X,C=c)$ and $P(C=c)$ from the data, assuming strong ignorability. While imbalance or not only leads to different levels of estimation errors of $P(Y|X,C=c)$ and $P(C=c)$, which should not affect the identification of $P(Y|do(X))$ itself with same identification and estimation process.
>
> I would be happy to learn any theorem in Pearl's book [b] which claims that **training a model on a balanced observational dataset with $P(C=1)=P(C=0)$ would lead to causal identification and unbiased estimate of an interventional distribution while training on an imbalanced dataset with $P(C=1)\ne P(C=0)$ cannot achieve the same**.
>
> > we can ensure that $P(C=1)=P(C=0)$, meaning that males and females are equally represented in both the treated and untreated groups.
>
> This claim is not correct. $P(C=1)=P(C=0)$ is a necessary but not sufficient condition for that claim males and females are equally represented in both the treated and untreated groups, which is equivalent to $P(C=1|X=x)=P(C=0|X=x), x=\{0,1\}$.
>
> BTW, Simpson's paradox is a fact or an observation that $P(Y=y|X)>P(Y=y'|X)$ but $P(Y=y|X,C)<P(Y=y'|X,C)$ can happen, it is not a theorem. So applying a fact to your example does not make much sense to me.

---

> > ### Author Response · Authors · 2024-11-29
> >
> > We appreciate the reviewers for engaging in an in-depth discussion with us, particularly for clarifying certain causal concepts.
> >
> > > that claim males and females are equally represented in both the treated and untreated groups, which is equivalent to $P(C=1|X=x)=P(C=0|X=x),x=0,1$
> >
> > Regarding your suggestion, should $P(C|X=1)$ be corrected to $P(X=1|C)$ to indicate that men and women have the same willingness to participate in the treatment? Based on the analysis of Simpson's paradox and propensity scores in [Pearl et al., 2016, p. 72, Sec. 3.6], we understand that $P(X|C)$ reflects the **propensity** for different genders to participate in the treatment. It is precisely because the **number** of women participating in the treatment is greater than that of men, i.e., $P(X=1|C=1) > P(X=1|C=0)$, that the causal arrow $C \rightarrow X$ is established. Moreover, estrogen suppresses the effectiveness of the treatment, leading to $C \rightarrow Y$. Combining these two relationships, we conclude that gender $C$ acts as a confounder, forming $X \leftarrow C \rightarrow Y$.
> >
> > We clarify our previously unreasonable statement $P(C=0)=P(C=1)$, which has been revised to $\mathbf{P(X|C=0)=P(X|C=1)}$. We will align the example of Simpson's paradox with our synthetic experiment to further illustrate the role of the confounder.
> >
> > In Simpson's paradox, the ratio of women to men participating in the treatment is 263:87, representing the difference in the **propensity probability** $P(X=1|C)$. In our synthetic data, this corresponds to a dataset where the number of images dominated by the head of an object versus those dominated by the body is 90:10, representing the higher propensity for head-dominated images to appear, i.e., $P(X|C=0) > P(X|C=1)$. Additionally, in Simpson's paradox, $C \rightarrow Y$ indicates that estrogen suppresses the treatment, which corresponds in our scenario to the imbalance in model parameters: OpenClip exhibits a preference in its predictive distribution for "the head of the object" (i.e., making more accurate predictions on test samples where the head of the object is not occluded).
> >
> > Returning to $P(Y|do(X))$, this represents the removal of the causal arrow from the confounder $C$ to $X$, indicating that the propensity relationship $\mathbf{P(X|C)}$ no longer holds. In other words, in the post-intervention world, the manipulated probability satisfies $P_m(X|C) = P_m(X)$, implying that $C \perp X$ after the intervention. Therefore, a randomized controlled trial can quantitatively estimate $P(Y|do(X))$, requiring only that $\mathbf{P(X|C=0) = P(X|C=1)}$.
> >
> > For example, in Simpson's paradox, equal numbers of men and women participating in treatment correspond, in our context, to an equal ratio of head-dominated and body-dominated images (1:1). In such a dataset, there is no propensity-driven occlusion of certain parts of the object; **instead, appearances occur randomly**. Achieving this in practice, however, is highly challenging. For instance, when collecting train images, most will feature the train's head while only a few include the train's body. Similarly, human photographic preferences tend to focus on the head of an animal (e.g., facial features, gaze) rather than its entire body when observing or photographing at close range.
> >
> > [Pearl et al., 2016] Pearl, Judea. Causal inference in statistics: a primer. John Wiley & Sons, 2016.
> >
> >
> > We sincerely appreciate your thoughtful and detailed feedback, which has significantly contributed to the improvement of our paper. If there are any further questions or clarifications needed, please do not hesitate to reach out. We would be more than happy to address any additional concerns or provide further explanations.

---

> > > ### Author Response · Authors · 2024-12-01
> > > **Warm Reminder from Authors**
> > >
> > > Dear Reviewer,
> > >
> > > Thank you for engaging in such an in-depth discussion with us. We greatly appreciate the opportunity to clarify the causal concepts behind our experiment and sincerely hope that our responses and the revised manuscript have addressed your concerns to your satisfaction.
> > >
> > > As the discussion period comes to a close, we would be truly grateful if you could share your thoughts on whether our efforts have sufficiently resolved your concerns. Your constructive feedback has been invaluable throughout this process, and we deeply respect your judgment. Should you have any remaining questions or require further clarification or additional experiments, we would be more than happy to provide them to ensure the highest quality of this work.
> > >
> > > Thank you once again for your time and thoughtful review.
> > >
> > > Best regards,
> > >
> > > The Author

---

> ### Comment · Reviewer_DbkM · 2024-12-01
>
> 1. I agree with your description of the example from [Pearl, 2016].
> 2. If $P(X=1|C=c)=P(X=0|C=c)$ for all $c$, then there is no causal effect of $C$ on $X$, leading to randomization. The key is, in this case, $P(Y|do(X))=P(Y|X)$, so you can estimate $P(Y|do(X))$ without using the backdoor criterion. Under this setting, you might be able to show there exists confounding bias by comparing $P(Y|X)$ in this setting and $P(Y|X)$ with $P(X=1|C=c) \ne P(X=0|C=c)$.

---

> ### Author Response · Authors · 2024-12-01
>
> Thanks for your in-depth discussion with us. We are very pleased that you agree with our description of the example. Regarding our experiments, we composed two training sets ($A_{train}$ and $B_{train}$) and two test sets ($A_{test}$ and $B_{test}$) .
>
> 1) For each incomplete semantic factor, 50 samples were generated. This dataset is referred to as $A_{train}$. ($P(X|C=0) = P(X|C=1)$)
>
> 2) For $C=0$, 90 samples were generated, and for $C=1$, 10 samples were generated. This dataset is referred to as $B_{train}$. ($P(X|C=0) \neq P(X|C=1)$)
>
> The model trained on $A_{{train}}$, when used for inference, can be interpreted as estimating $P(Y \mid {do}(X))$, since $P(X|C=0) = P(X|C=1)$. The model trained on $B_{{train}}$, when used for inference, can be interpreted as estimating $P(Y \mid X)$, since $P(X|C=0) \neq P(X|C=1)$.
>
> For evaluation, the test datasets are constructed under the following conditions:
>
> 1) For each incomplete semantic factor, 50 samples were generated. This dataset is referred to as $A_{test}$.
>
> 2) For $C=0$, 90 samples were generated, and for $C=1$, 10 samples were generated. This dataset is referred to as $B_{test}$.
>
> We conducted four groups of experiments, as outlined below:
>
> a) The model is trained on $A_{{train}}$ and evaluate on $A_{{test}}$.
>
> b) The model is trained on $A_{{train}}$ and evaluate on $B_{{test}}$.
>
> c) The model is trained on $B_{{train}}$ and evaluate on $A_{{test}}$.
>
> d) The model is trained on $B_{{train}}$ and evaluate on $B_{{test}}$.
>
> As shown in Fig.12 in the revised paper, we observe that the model trained on $A_{{train}}$ performs similarly on both $A_{{test}}$ and $B_{{test}}$. This result indicates that the trained model does not learn the correlation.
>
> However, when the model is trained on $B_{{train}}$, we observe that the model trained on $B_{{train}}$ exhibits different performance between $A_{{test}}$ and $B_{{test}}$. Considering that if the trained model is not affected by incomplete semantic factors, it should present the same performance between $A_{{test}}$ and $B_{{test}}$. However, we find the evaluation results are different, verifying the confounder's influence.
>
> >Under this setting, you might be able to show there exists confounding bias by comparing  $P(Y|X)$ in this setting and  P(Y|X) with $P(X=1|C=c) \neq P(X=0|C=c)$.
>
> Moreover, when evaluated on the $A_{{test}}$, we find that the model trained on $A_{{train}}$ achieves better performance compared with the model trained on $B_{{train}}$. The reason is we constitute $A_{{train}}$ following $P(X=1|C=c) = P(X=0|C=c)$ but $B_{{train}}$ following $P(X=1|C=c) \neq P(X=0|C=c)$. By these experiments, we show ***"there exists confounding bias by comparing  $P(Y|X)$ in this setting and  P(Y|X) with $P(X=1|C=c) \neq P(X=0|C=c)$."***
>
> Thank you once again for your time and insightful review. We do believe your valuable comments are very important to us. If you have any further questions or require additional clarification or experiments, please don't hesitate to reach out. We would be more than happy to provide whatever is needed to ensure the highest quality of this work.

---

> > ### Author Response · Authors · 2024-12-02
> >
> > Dear Reviewer DbkM,
> >
> > We sincerely thank you for the recognition of our work and for engaging in an in-depth discussion with us. We greatly appreciate the time and effort invested, as well as the insightful comments provided. We commit to incorporating the discussed points into our paper to enhance its quality. Once again, thank you for your valuable feedback and thoughtful contributions.
> >
> > Best regards,
> >
> > The Author

---

### Official Review · Reviewer_cz4B · 2024-11-04

**Soundness:** 3
**Presentation:** 4
**Contribution:** 4
**Rating:** 8
**Confidence:** 3

**Summary:**

In the context of foundation model and PEFT, this paper defines two imbalance terms for long-tailed data, i.e., data-imbalance and parameter imbalance. The paper argues that only considering data imbalance is insufficient for PEFT on foundation models, and proposes an effective strategy to address the problem of parameter imbalance from a causal graph perspective. Numerical experiments have been conducted to support the arguments, and their proposed strategies.

The method's lack of efficiency is analyzed and discussed.

**Strengths:**

1. [Contribution] The proposed argument is important for the community, which is a generally existing problem while rarely mentioned. The propose of this problem is imperative.
2. [Soundness] The authors conduct a large number of experiments on SOTA models and popular long-tail datasets to demonstrate their arguments.
3. [Presentation] This paper is well organized. The step-by-step exploration of the problem provides an intriguing reading experience.

**Weaknesses:**

Although the introduced problem is important, the solution is kind of trivial. The simple combination of different models could partially resolve the problem, while each of the foundation models contains a large number of parameters, which is not applicable.

**Questions:**

1. In Fig. 3, how is the right picture plotted? How is the parameter imbalance computed for each sample?
2. In equation. 7, how is $P(Y=y|bc)$ implemented in your experiments. Do you run each model once and generate a $P(Y=y|bc)$? and how these models are jointly trained (or fine-tuned)?
3. In lines 184-195, it mentioned that "$P_P(Y)$ is inaccessible and we cannot use Eq. (1)". How is $P_P(Y)$  related to Eq. (1) where the symbol is not mentioned.

---

> ### Author Response · Authors · 2024-11-20
>
> We sincerely appreciate your thoughtful feedback and constructive suggestions. To address your concerns comprehensively, we have carefully reviewed each point you raised. In the following, your weaknesses and questions are summarized in quotes, followed by our point-by-point responses.
>
> **Weakness 1**
> >Although the introduced problem is important, the solution is kind of trivial. The simple combination of different models could partially resolve the problem, while each of the foundation models contains a large number of parameters, which is not applicable.
>
> **Re:**
> Thanks for your advice. As shown below, larger values of $M$ allow for the discovery of more incomplete semantic factors, but they also lead to significantly higher computational expenses. This trade-off highlights the importance of selecting an optimal $M$ that balances performance gains and computational efficiency.
>
> |  | FLOPs |
> |--|--|
> |M=1|239.06|
> |M=2|478.13|
> |M=3|717.19|
>
> Our method is built on parameter-efficient fine-tuning and we need to store the parameters of multiple foundation models (CLIP, OpenCLIP, MetaCLIP). However, for different downstream tasks (ImageNet-LT, Places365-LT, iNaturalist2018), we only retain a single copy of the foundation model parameters and store additional task-specific parameters for different downstream tasks separately. Since the task-specific parameters require minimal storage, the additional costs introduced by our approach are relatively negligible. We have provided a detailed discussion of this limitation in the revised paper.
>
> **Question 1**
> >In Fig. 3, how is the right picture plotted? How is the parameter imbalance computed for each sample?
>
> **Re:**
> The x-axis in the left image of Fig.3 represents the class indices sorted according to the downstream label prior $P_S(Y)$ while the x-axis in the right image is sorted based on the estimated upstream label prior $\widehat{P}_P(Y)$. Since the pre-training data is inaccessible, we adopt the method proposed in [a] to estimate the upstream label prior $\widehat{P}_P(Y)$.
>
> **Question 2**
> >In equation. 7, how is $P(Y=y|b, c)$ implemented in your experiments. Do you run each model once and generate a $P(Y=y|b, c)$ and how these models are jointly trained (or fine-tuned)?
>
> **Re:**
> Yes, we run each model once and generate $P(Y=y|b, c)$. During the inference, we use our backdoor adjustment to ensemble different fine-tuned models (incomplete semantic factors) and make final predictions.
>
> We also experiment with jointly training multiple foundation models to address the influence of incomplete semantic factors during the training phase. For details, we ensemble different fine-tuned models during the training phase. However, as shown in our results, this approach did not yield significant performance improvements. Joint training did not provide substantial additional benefits compared to our proposed method.
> |  | Places365-LT |ImageNet-LT|
> |--|--|--|
> |Jointly|52.30|78.15|
> |Ours|53.01| 79.57|
>
> **Question 3**
> >In lines 184-195, it mentioned that " $P_P(Y)$ is inaccessible and we cannot use Eq. (1)". How is $P_P(Y)$ related to Eq. (1) where the symbol is not mentioned.
>
> **Re:**
> We denote the $P_P(Y)$ as the label prior of the pre-training data to train the foundation model, and $\widehat{P}_P(Y)$ as the estimated label prior via GLA. Eq.1 shows how to bridge the gap between source distribution $S$ and target distribution $T$. Therefore, when we adapt this to the pre-trained distribution $P$, the equation should be changed as follows.
>
> $P_{T}(Y=y \mid x) = \frac{P_{T}(x \mid Y=y) P_{T}(Y=y)}{P_{T}(x)}  \propto P_{P}(x \mid Y=y)P_{T}(Y=y)  \propto \frac{P_{T}(Y=y)}{P_{P}(Y=y)}P_{P}(Y=y \mid x)$
>
>
> [a] Beier Zhu, Kaihua Tang, Qianru Sun, and Hanwang Zhang. Generalized logit adjustment: Calibrating fine-tuned models by removing label bias in foundation models. Advances in Neural Information Processing Systems, 36, 2024.

---

> > ### Comment · Reviewer_cz4B · 2024-11-27
> >
> > I appreciate the authors' response. The new experimental results have largely addressed my concern. However. the efficiency of the proposed method is still an issue. Considering the authors have mentioned this limitation in their manuscript, I'll retain my original score.

---

> > > ### Author Response · Authors · 2024-11-27
> > >
> > > We sincerely appreciate your recognition of our work! This process has provided us with valuable insights, and it has been an honor to benefit from your time and expertise. Your feedback has been instrumental in helping us refine and enhance our work.

---

### Official Review · Reviewer_JGAh · 2024-11-04

**Soundness:** 3
**Presentation:** 3
**Contribution:** 3
**Rating:** 5
**Confidence:** 4

**Summary:**

This paper addresses the long-tail learning problem in foundation model fine-tuning from a novel perspective by considering both parameter imbalance (from pre-training) and data imbalance (from downstream tasks). The authors propose a causal inference framework to analyze these two types of imbalance and introduce a backdoor adjustment method to learn the true causal effects. Experiments conducted on three long-tailed datasets (ImageNet-LT, Places365-LT, iNaturalist2018) with various foundation models demonstrate the effectiveness of the proposed approach.

**Strengths:**

The paper presents a novel perspective on long-tail learning by considering the dual impact of pre-training and fine-tuning imbalance, which has been largely overlooked in previous work. The causal inference framework provides a principled way to analyze and address these challenges.

**Weaknesses:**

1. The definition and quantification of "parameter imbalance" lack rigorous theoretical justification. The causal relationship between pre-training data imbalance and parameter imbalance is not thoroughly explained. The selection of semantic factors as confounders needs stronger theoretical support.

2. The claim that parameter imbalance has greater impact than data imbalance needs more empirical evidence.

3. Incomplete ablation studies for key components and lack of analysis on computational overhead and efficiency.

**Questions:**

1. How do you quantify parameter imbalance without access to the pre-training data? What metrics are used?

2. Can you provide theoretical justification for why semantic factors should be treated as confounders in the causal framework?

3. What is the computational overhead of the proposed method compared to standard fine-tuning?

4. How does the method perform when the degree of imbalance varies between pre-training and fine-tuning?

---

> ### Author Response · Authors · 2024-11-20
>
> Thanks for your valuable comments and suggestions! We have carefully addressed all your weaknesses and questions. In the following, your weaknesses and questions are summarized in quotes, followed by our point-by-point responses.
>
> **Weakness 1**
> > The definition and quantification of "parameter imbalance" lack rigorous theoretical justification. The causal relationship between pre-training data imbalance and parameter imbalance is not thoroughly explained. The selection of semantic factors as confounders needs stronger theoretical support.
>
> **Re:**
>
> ***(1)Theoretical justification.*** Thank you for pointing this out. In the revised paper Sec.4.1, we have added a theoretical definition of parameter imbalance to help readers better understand the concept. Briefly, we use the imbalanced factor (IF) to quantify the extent to which foundation models are influenced by different pre-training datasets.
>
> ***(2)Causal relationship between pre-training data and parameter imbalance.*** Since the pre-training data of the foundation model is inaccessible, its influence is primarily reflected in the pre-trained weights or parameters, which we refer to as parameter imbalance. Consequently, we describe the imbalance in pre-training data as the parameter imbalance when adapting the model to downstream tasks. In the revised paper, we also provide a theoretical justification for parameter imbalance to help readers better understand this concept.
>
> ***(3)Confounder.*** We imply that **the incomplete semantic factor** $C$, caused by parameter imbalance $D$, serves as a confounder. We will provide theoretical support from two aspects: 1) the theoretical definition of confounders, and 2) the spurious correlation induced by the backdoor path $X \leftarrow C \rightarrow Y$.
>
> ***(3.1)Theoretical definition of confounders.*** When a third variable $Z$ influences both $X$ and $Y$, we say that $X$ and $Y$ are confounded. Such a variable $Z$ is referred to as a confounder of $X$ and $Y$ [Pearl et al., 2009, p. 55, Def. 2.7.1 & p. 184, Sec. 6.22]. Specifically, it satisfies the fork structure $X \leftarrow Z \rightarrow Y$. This latent common cause $Z$ can create a spurious correlation between $X$ and $Y$, making the observed statistical relationship $P(Y|X)$ potentially misleading. To address this, the causal relationship $P(Y|\text{do}(X))$ is used as a replacement for the correlation $P(Y|X)$.
>
> [Pearl et al., 2009] Pearl, Judea. *Causality*. Cambridge university press, 2009.
>
> ***(3.2) Why are the incomplete semantic factors derived from a parameter imbalance model confounders.*** The example in the first row in Fig.4 explains why the parameter-imbalanced model $D$ leads to an **incomplete semantic factor** $C$. For example, the semantic factor available to OpenCLIP is in the head of the object, while the semantic factor available to MetaCLIP is in the body of the object. The relationship between the confounding path $X \leftarrow C \rightarrow Y$ is unstable. When the training set model predicts the label with the help of the body semantics of the object, if the body of a sample of a certain class in the test example is occluded, the model will give an incorrect prediction. From this example, it can be seen that an ideal model needs to learn the causal relationship $P(Y|do(X))$ between the input and the label, rather than fitting the unstable confounding path $X \leftarrow C \rightarrow Y$, in order to achieve generalization between different distributions.

---

> ### Author Response · Authors · 2024-11-20
>
> **Weakness 2**
> > The claim that parameter imbalance has greater impact than data imbalance needs more empirical evidence.
>
> **Re:**
> In Fig. 3, the class indices in the left image are sorted based on the label prior of the downstream data, while the class indices in the right image are sorted based on the estimated label prior of the pre-training data. (Since the pre-training data is inaccessible, we use GLA [a] to estimate the label prior of the pre-training data.) After applying Logit Adjustment, we observe that the downward trend in the performance curve is mitigated, indicating that data imbalance can be effectively addressed. However, the right image still exhibits a downward trend, suggesting that parameter imbalance persists. Given that data imbalance can be alleviated while parameter imbalance remains unresolved, we conclude that parameter imbalance has a greater impact than data imbalance.
>
> To further support this conclusion, we train a model from scratch on ImageNet-LT using ResNet-50. Since the parameters of ResNet-50 are randomly initialized, the trained model is only influenced by data imbalance. After training, we calculate the accuracy of each class and sort the classes based on the estimated label prior of the pre-training data, $\widehat{P}_P(Y)$. Because parameter imbalance does not affect the trained model in this setup, the performance curve after sorting should not display a downward trend. As shown in Fig.9, this result validates our argument. Thus, in this comparative experiment, the downward trend observed in the right image of Fig.3 is attributable to the imbalance in the pre-training data, further verifying the impact of parameter imbalance.
>
> **Weakness 3**
> >Incomplete ablation studies for key components and lack of analysis on computational overhead and efficiency.
>
> **Re:**
> Thank you for your advice. The key component of our method is the hyperparameter $M$. As shown in Tab.8, our experiments demonstrate that increasing $M$ improves performance. However, as shown in the following, larger values of $M$ also result in higher computational costs, measured in FLOPs. This trade-off underscores the importance of selecting an optimal $M$ to balance performance gains and computational efficiency.
>
> |  | FLOPs |
> |--|--|
> |M=1|239.06|
> |M=2|478.13|
> |M=3|717.19|
>
> **Question 1**
> >How do you quantify parameter imbalance without access to the pre-training data? What metrics are used?
>
> **Re:**
> Although we cannot access the pre-training data directly, we can use a validation set to estimate the label frequency indirectly. Following GLA [a], we estimate the label frequency for different datasets. Therefore, we can also use the label frequency and imbalanced factor (IF) as the metric to evaluate parameter imbalance.

---

> ### Author Response · Authors · 2024-11-20
>
> **Question 2**
> >Can you provide theoretical justification for why semantic factors should be treated as confounders in the causal framework?
>
> **Re:**
> As mentioned in Weakness 1, we first provide the definition of confounders in the causal framework: When a third variable $Z$ influences both $X$ and $Y$, we say that $X$ and $Y$ are confounded. Such a variable $Z$ is referred to as a confounder of $X$ and $Y$ [Pearl et al., 2009, p. 55, Def. 2.7.1 & p. 184, Sec. 6.22]. Specifically, it satisfies the fork structure $X \leftarrow Z \rightarrow Y$. This latent common cause $Z$ can create a spurious correlation between $X$ and $Y$, making the observed statistical relationship $P(Y|X)$ potentially misleading. To address this, the causal relationship $P(Y|\text{do}(X))$ is used as a replacement for the correlation $P(Y|X)$.
>
> Secondly, we explain why the incomplete semantic factor $C$ in our problem setting can create a backdoor path $X \leftarrow C \rightarrow Y$, leading to a spurious correlation. The example in the first row in Fig.4 explains why the parameter-imbalanced model $D$ leads to an **incomplete semantic factor** $C$. For example, the semantic factor available to OpenCLIP is in the head of the object, while the semantic factor available to MetaCLIP is in the body of the object.
>
> We emphasize that the semantic factors obtained due to model imbalance are incomplete. Therefore, for rigor, we have modified the causal diagram in Fig. 5 to include the inaccessible semantic factor $U$. From the perspective of data generation, $C$ and $U$ together constitute $X$, thus $X \leftarrow C$ holds. On the other hand, the path $C \rightarrow Y$ indicates that the predicted label distributions follow their own parameter imbalance. A parameter-imbalanced model, OpenCLIP, consistently predicts a "head of the object" label distribution (i.e., making more accurate predictions on test samples where the head of the object is not occluded), regardless of the test distributions. Similarly, the parameter-imbalanced model MetaCLIP exhibits a distinct "body of the object" preference across different test distributions.
>
> The relationship between the confounding path $X \leftarrow C \rightarrow Y$ is unstable. When the training set model predicts the label with the help of the body semantics of the object, if the body of a sample of a certain class in the test example is occluded, the model will give an incorrect prediction. From this example, it can be seen that an ideal model needs to learn the causal relationship $P(Y|do(X))$ between the input and the label, rather than fitting the unstable confounding path $X \leftarrow C \rightarrow Y$, in order to achieve generalization between different distributions.
>
> **Question 3**
> >What is the computational overhead of the proposed method compared to standard fine-tuning?
>
> **Re:**
> See the response for the Weakness 3
>
> **Question 4**
> >How does the method perform when the degree of imbalance varies between pre-training and fine-tuning?
>
> **Re:**
> Thank you for your insightful advice. In our paper, we address both data imbalance and parameter imbalance simultaneously, with analyses primarily based on commonly used datasets. To better understand the impact of parameter imbalance, we consider the following scenarios: (1) **Consistency**: The data imbalance and parameter imbalance are consistent, where the head classes grouped by the data imbalance are also the head classes grouped by the parameter imbalance. (2) **Reverse**: The data imbalance and parameter imbalance are reversed, where the head classes grouped by the data imbalance are the tail classes grouped by the parameter imbalance. (3)**Balance**: The downstream data is balanced. To simulate these scenarios, we construct additional imbalanced datasets by downsampling the Places365 dataset, as illustrated in Fig. 7. We evaluate performance using the Places365-LT balanced test set. To reduce the influence of randomness, we create three datasets for each scenario and report the average performance across them.
>
> The results, shown in Tab. 13, indicate that when the downstream data is balanced, the model achieves the best performance, as this setup aligns most closely with the distribution of the downstream task. Interestingly, the **Reverse** scenario outperforms the **Consistency** scenario. In the Reverse setting, the tail classes (in terms of parameter imbalance) can be viewed as head classes (in terms of data imbalance), suggesting that parameter imbalance can be mitigated by increasing the number of training samples for tail classes. However, in real-world applications, collecting sufficient data for tail classes is often challenging, and simply adding more data is not a practical solution to this issue.
>
> [a] Beier Zhu, Kaihua Tang, Qianru Sun, and Hanwang Zhang. Generalized logit adjustment: Calibrating fine-tuned models by removing label bias in foundation models. Advances in Neural Information Processing Systems, 36, 2024.

---

> > ### Author Response · Authors · 2024-11-25
> >
> > Dear Reviewer,
> >
> > Thank you once again for your professional and insightful feedback. We sincerely value your suggestion to define the parameter imbalance and analyze the computational overhead and efficiency. We have incorporated these aspects into our revisions and hope they address all of your concerns.
> >
> > As the discussion period is nearing its conclusion, we would be grateful if you could let us know whether our responses and the revised manuscript sufficiently resolve your questions or if you have any further comments or suggestions.
> >
> > Best regards,
> > The Authors

---

> > > ### Comment · Reviewer_JGAh · 2024-11-25
> > >
> > > Thank you very much for the author's response, which has addressed some of my questions regarding the definitions. However, I still believe that this paper's contribution to the field of long-tail learning is limited. Overall, after considering the author's reply and the feedback from other reviewers, I feel that this paper still requires further improvement, so I will maintain my current score for now.

---

> > > > ### Author Response · Authors · 2024-11-25
> > > >
> > > > Thanks for your reply. We do believe that our work makes a significant contribution to the long-tailed field. Compared to previous studies, our work is the first to explicitly address the issue of pre-training bias under long-tailed distribution. The main contributions of our study are summarized as follows:
> > > >
> > > > 1.	We address a practical problem where both the data used for training the foundation model and for adaptation to downstream tasks are imbalanced.
> > > >
> > > > 2.	We perform detailed experiments to analyze the impact of pre-training bias under long-tailed distribution, which has not been explored before.
> > > >
> > > > 3.	We identify that incomplete semantic factors encourage the model to learn spurious correlations between input samples and labels, thereby limiting its generalization ability. To tackle this, we construct a causal graph and propose a backdoor adjustment method to mitigate the negative impact of confounders.
> > > >
> > > > Regarding your previous suggestions, we have carefully addressed each of them and improved our revised paper. Specifically:
> > > >
> > > > On the issue of parameter imbalance, we have added explanations in the Sec.4.1 of the revised PDF.
> > > >
> > > > On the issue of the great impact of parameter imbalance, we have added experiments in the Sec.D.5 of the revised PDF.
> > > >
> > > > On the issue of computational efficiency, we have added experiments in the Sec.6.3 of the revised PDF.
> > > >
> > > > On the issue of the theoretical justification of incomplete semantic factor, we have added explanations in the Sec.B and Sec.D.4 of the revised PDF.
> > > >
> > > > On the issue of the degree of imbalance varies between pre-training and fine-tuning, we have added experiments in the Sec.D.3 of the revised PDF.
> > > >
> > > > We have revised and refined the manuscript in accordance with your feedback. Should you have any further questions or concerns about our work, please do not hesitate to contact us.

---

> > > > > ### Author Response · Authors · 2024-11-27
> > > > >
> > > > > Thank you for your time and effort. We have reiterated our contributions in detail. As the deadline for uploading the revised PDF is approaching, could you kindly let us know where our paper could be further improved? If you have any questions or concerns, we would be happy to provide detailed responses and carefully revise our paper based on your suggestions.

---

> > > > > > ### Author Response · Authors · 2024-12-02
> > > > > >
> > > > > > Dear Reviewer JGAh,
> > > > > >
> > > > > > Thank you once again for your time and thoughtful feedback on our work. We greatly appreciate the effort you have dedicated to reviewing our submission.
> > > > > >
> > > > > > At this stage, many reviewers have expressed positive evaluations of the paper. Following extensive discussions and revisions, we believe our manuscript has been significantly improved, with many concerns now addressed. In light of these updates and clarifications, we kindly hope you might consider revisiting your assessment.
> > > > > >
> > > > > > Should you have any further questions or concerns, we would be more than happy to provide detailed responses and make any necessary refinements to the paper based on your insights.
> > > > > >
> > > > > > Thank you again for your invaluable input.
> > > > > >
> > > > > > Best regards,
> > > > > >
> > > > > > The Authors

---

### Author Response · Authors · 2024-11-24
**Warm Reminder from Authors**

Dear Reviewers,

We sincerely thank you once again for your professional and constructive feedback. We have provided detailed responses regarding the causal graph and our experimental settings, and we hope these have addressed all of your concerns.

As the discussion period is nearing its conclusion, we would greatly appreciate it if you could let us know whether our responses and the revised manuscript have adequately addressed your concerns, or if there are any additional comments.

Sincerely,
The Authors

---

### Meta-Review · Area_Chair_Q64U · 2024-12-15

**Metareview:**

This paper investigates long-tail learning in foundation models when the pre-trained model was trained on imbalanced data. The authors show that the biased model cannot easily be corrected with a balanced downstream dataset and propose a backdoor adjustment to address this bias.

Overall, this is a complete set of experiments and discussion, first showing that parameter imbalance cannot be resolved with GLA, and proposing a novel correction. Experiments are conducted on 3 datasets, multiple baselines are considered and abaltion studies performed. However, the reviewers pointed to a lack of formalization of parameter imbalance, lack of theoretical justification for some of their design choices (e.g. $C$ as confounder) and a lmited contribution to the field of long-tail learning.

Having reviewed all the comments as well as the authors responses, I believe that some of the concerns remain and hence recommend rejection at this time. I also found that the most positivee review lacked depth and have not weighted it heavily in my decision.

**Additional Comments On Reviewer Discussion:**

The authors engaged in the discussion, providing detailed responses to the concerns of the reviewers as well as a revised version of their manuscript. The reviewers have all acknowledged this response but felt that some of their concerns remained, especially in terms of contribution and theoretical work. This was confirmed during the reviewer discussion.

---

### Decision · Program_Chairs · 2025-01-22

Reject